# Curvature domains in V4 of macaque monkey

Jia Ming Hu[1], Xue Mei Song[1,2], Qiannan Wang[1], Anna Wang Roe[1,2,3]*

[1]Department of Neurology of the Second Affiliated Hospital, Interdisciplinary Institute of Neuroscience and Technology, School of Medicine, Zhejiang University, Hangzhou, China; [2]Key Laboratory for Biomedical Engineering, of Ministry of Education, Zhejiang University, Hangzhou, China; [3]Division of Neuroscience, Oregon National Primate Research Center, Oregon Health & Science University, Beaverton, United States

**Abstract** An important aspect of visual object recognition is the ability to perceive object shape. Two basic components of complex shapes are straight and curved contours. A large body of evidence suggests a modular hierarchy for shape representation progressing from simple and complex orientation in early areas V1 and V2, to increasingly complex stages of curvature representation in V4, TEO, and TE. Here, we reinforce and extend the concept of modular representation. Using intrinsic signal optical imaging in Macaque area V4, we find sub-millimeter sized modules for curvature representation that are organized from low to high curvatures as well as domains with complex curvature preference. We propose a possible 'curvature hypercolumn' within V4. In combination with previous studies, we suggest that the key emergent functions at each stage of cortical processing are represented in systematic, modular maps.

## Introduction

Recognizing the shapes of objects requires information about local contour features, such as orientation and curvature. The encoding of contour orientation in the visual system begins in primary visual cortex (V1) with neurons selective for the orientation of visual contours, and, in the second visual area (V2), with neurons selective for cue-invariant contour orientation (that is, independent of whether contours are defined by luminance, motion, color, or depth cues). These two functions are first embodied in the '*orientation domain*' in V1 (*Hubel and Wiesel, 1968*; *Blasdel and Salama, 1986*; *Grinvald et al., 1986*) and the '*higher order orientation domain*' in V2 (*Ramsden et al., 2001*; *Chen et al., 2016*). These signature domains in V1 and V2 mark initial computational elements of shape representation.

We investigate whether there are functional domains representing contour curvature in area V4 of macaque monkey visual cortex, an important intermediate stage of shape encoding. Although both V2 (*Hegdé and Van Essen, 2003*; *Ito and Komatsu, 2004*; *Anzai et al., 2007*) and V4 are considered intermediate stages of shape representation, a greater body of neural (*Pasupathy and Connor, 2001*; *Hegdé and Van Essen, 2007*; *Nandy et al., 2016*) and computational modeling (*Murphy and Finkel, 2007*; *Pospisil et al., 2018*; *Wei et al., 2018*) evidence points to area V4 as a locus of curvature information processing. V4 neurons are responsive to contour curvature and can be selective for degree of curvature and orientation of curvature (*Pasupathy and Connor, 2001*; *Nandy et al., 2016*). However, it is unknown whether such neuronal responses are organized in any way akin to orientation maps in V1 and V2. Previous studies have shown the functional organization of V4 comprises alternating bands of 'orientation' and 'color' preference (*Tanigawa et al., 2010*; *Li et al., 2013*; *Ghose and Ts'o, 2017*), as well as organization for disparity defined orientation (*Fang et al., 2019*), motion direction (*Li et al., 2013*), and spatial frequency

*For correspondence:
annawang@zju.edu.cn

**Competing interests:** The authors declare that no competing interests exist.

(*Lu et al., 2018*), but maps for curvature organization have not been demonstrated. Here, we hypothesized that orientation bands in V4 are regions of shape representation that include both orientation and curvature domains.

We used intrinsic signal optical imaging, a method that is well-suited for studying functional domain responses, to examine curvature response in monkey V4. We hypothesized that curvature domains would meet the following criteria: (1) *Band location in V4:* We predicted that curvature domains would be located within the' orientation bands' and not the 'color' bands. (2) *Distinct response preference*: We expected that curvature domains should prefer curved stimuli over straight stimuli, and, furthermore, would exhibit selectivity for different degrees of curvature (from low to high) or different orientations of a curve (from 0˚ to 360˚). (2) *Curvature maps*: Just as orientation and color domains have distinct maps, we predicted that curvature domains should have systematic representation for curvature degree and curvature orientation. (3) *Response consistency*: We also predicted that curvature domains would exhibit similar response to curved gratings and curved lines of similar curvature. (4) *Hypercolumn*: Finally, we sought evidence for the possibility of a curvature hypercolumn within V4. Our data suggest that there is a systematic 'curvature map' within V4. While curvature domains have previously been identified within V4 (*Yue et al., 2014*), this study shows such systematic maps at domain scale.

## Results

### General approach

Our approach is to understand the functional organization of curvature response in V4. We approached this study from a global to local scale in V4 by testing several predictions. First, we hypothesized that, as part of the shape encoding network, curvature domains would fall within the V4 'orientation bands' rather than the 'color bands'. Furthermore, within the orientation bands, curvature domains should be spatially distinct from the straight orientation domains. Second, analogous to the organization of orientation domains in V1, we predicted that curvature maps would represent curvature degree and curvature orientation in a systematic way. Third, if response is truly selective for curvature, curvature domains should exhibit the same selectivity for curvature content despite differences in detail (e.g. curved gratings vs curved lines). Fourth, we probed the hypothesis that a curvature hypercolumn exists in V4. This concept predicts that a local (e.g. ~1 degree) region of V4 should represent a range of curvature degrees and orientations. We present our findings below and provide a proposal for a curvature hypercolumn in V4.

### Curvature domains exist

Using intrinsic optical imaging, we imaged V4 in three hemispheres of two anesthetized macaque monkeys (27 sessions). In addition to conventional straight gratings, we designed curvature stimuli composed of sinusoidal curved gratings (*Figure 1*) with curvatures ranging from low to high. This stimulus (4 deg in size) comprises a central region (~1 deg) which contains the primary curvature content, while the flanking regions are relatively straight. Below, we will provide evidence that curvature response is attributed to this central region. As shown in *Table 1*, almost all experiments described in this study were conducted in at least two cases. Consistent with previous studies (*Tanigawa et al., 2010*; *Li et al., 2013*; *Ghose and Ts'o, 2017*), alternating regions (bands) of color preference vs. orientation preference were observed in V4, with small regions of overlap between bands (*Figure 2— figure supplement 1*). To precisely place our visual stimuli, we also mapped the retinotopy of V4 cortex using 0.2˚ horizontal and vertical lines and placed the center of visual stimuli (4˚ in size) on the monitor (*Figure 2—figure supplement 2*, see Materials and methods). We then zoomed in and obtained functional maps of V4. We used oriented achromatic and isoluminant color gratings to obtain maps for orientation (*Figure 2B*), color (*Figure 2D*), and high vs. low spatial frequency (*Figure 2E*). For all maps, the locations of functional domains were determined by t-value maps (t-map, two-tailed t test, p<0.01) which were calculated by comparing, pixel by pixel, the responses between two different conditions. Timecourses were examined to compare magnitudes of response of statistically significant domains.

We then examined response in V4 to straight vs. curved stimuli, consisting of moving curved (*Ponce et al., 2017*) sinusoidal gratings (four straight orientations: 0˚, 45˚, 90˚, 135˚; four curved

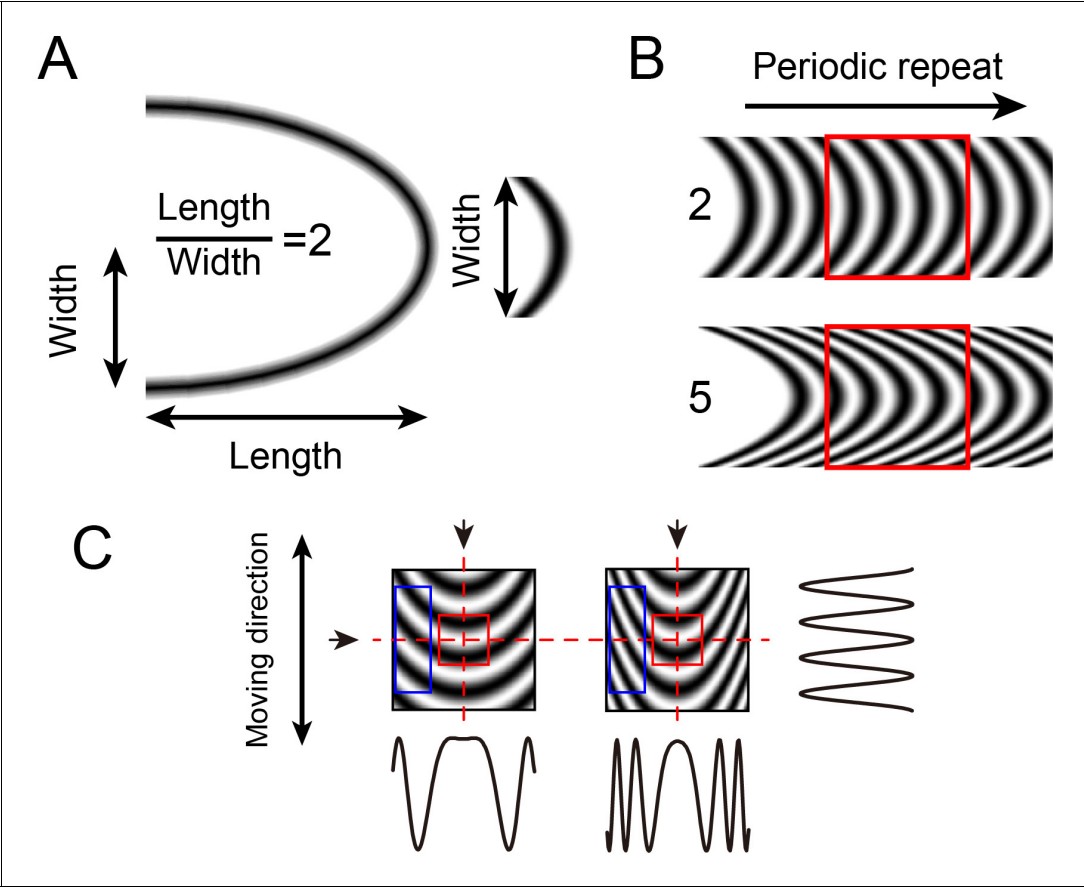

**Figure 1.** Curvature grating stimuli. (**A**) Calculation of curvature index. Width is half of the ellipse width. (**B**) Constructing curvature grating by periodic repeat. One low (index = 2) and one high (index = 5) curvature grating shown. (**C**) Luminance profiles of different stimuli in different directions. The profiles at the vertical axis (arrows at top, profile at right) of the two stimuli are similar, but differ in the horizontal axis (arrow at left, profiles below). The higher the curvature degree, the higher the spatial frequency towards the edge of the stimulus. Red square: central region, blue rectangle: flanking regions.

orientations: curved up, curved down, curved left, curved right), and flashed single curved (curved up, curved down) or straight lines (horizontal, vertical) (see insets in *Figure 2J and K*). To examine preference for curvature over straight gratings, we subtracted curvature (sum of all four curvature grating maps) minus straight (sum of all four straight grating maps), revealing dark (curvature preferring) domains (*Figure 2C*, two-tailed t test, p<0.01). These curvature maps were distinct from (straight) orientation maps (overlay of curvature and straight shown in *Figure 2I*). For simplicity of terminology, we will refer to these curvature vs. straight preferring domains as 'curvature domains'.

If these domains are indeed curvature domains, we predicted that they should be located within the 'orientation bands' and should be spatially distinct from straight orientation maps. We found that curvature domains (*Figure 2G–I*, red pixels) were distinct from color domains (*Figure 2G*, green pixels); they were also distinct from high spatial frequency domains (*Figure 2H*, green pixels), suggesting that these curvature responses are not simply due to the high spatial frequency components of the curved grating. Note also that these curvature domains have little overlap with low spatial frequency domains (*Figure 2—figure supplement 3*). In addition, as predicted, the curvature domains fell largely within the orientation band and were in close proximity to the straight orientation domains (*Figure 2I*, green pixels). As shown by the yellow pixels (locations of overlap), there is limited overlap between color and curvature domains (overlay in *Figure 2G*, Case 1, 4.9% overlap; all three cases, 5.7% overlap) and between high spatial frequency and curvature domains (overlay in *Figure 2H*, Case 1, 26.1% overlap; all three cases, 13.9% overlap). However, there is much greater overlap between curvature and orientation domains (*Figure 2I*, Case 1, 54.9% overlap; all three cases, 43.0% overlap), raising the possibility that curvature and orientation domains are different

**Table 1.** Case list.

| | | Case 1 | Case 2 | Case 3 |
|---|---|---|---|---|
| Curvature vs. straight map | Three cases (from three hemispheres of two animals) | *Figure 2C* and *Figure 2—figure supplement 5B* | *Figure 2—figure supplement 4F,G* and *Figure 2—figure supplement 5C,D* | *Figure 2—figure supplement 6B* |
| Curvature domain mask | Three cases | *Figure 2G* red pixels | *Figure 5A* right panel | *Figure 6D* |
| Curvature degree map | Three cases | *Figure 3* | *Figure 3—figure supplement 2A and B* | *Figure 7—figure supplement 1* |
| Curvature orientation map | Three cases | *Figure 3—figure supplement 2C,D* | *Figure 4* and 6A | *Figure 6D* |
| Response consistency (curvature orientation) | Two cases (from two hemispheres of two animals) | NA | *Figure 6A–C* | *Figure 6D–F* |
| Scrambled response | One case | NA | NA | *Figure 2—figure supplement 6* |
| Data used for response amplitude calculation | Cases 1 and 3 are from the same monkey. Response to color (Case 1: 50 trials; Case 2: 70 trials; Case 3: 50 trials with two different orientations) Response to high SF (Case 1: 80 trials; Case 2: 120 trials; Case 3: 100 trials with four different orientations) Response to 0, 45, 90, 135 (Case 1: 30 trials; Case 2: 30 trials; Case 3: 30 trials with the corresponding optimal orientations) Response to curvature grating (Case 1: 240 trials; Case 2: 240 trials; Case 3: 240 trials with four different curvature orientations and two different curvature degrees, a/b ratio 2 and 5) Response to flashed curvature (Case 2: 60 trials with two different curvature orientations; Case 3: 120 trials with four different curvature orientations) Number of pixels related to color domain (Case 1: 34986; Case 2: 10886; Case 3: 22560) Number of pixels related to high SF domain (Case 1: 31,480; Case 2: 10,687; Case 3: 29,757) Number of pixels related to 0, 45, 90, 135 orientation domain (Case 1: 33,676, 47,501, 51,798, 43,837; Case 2: 8225, 10,243, 4712, 8891; Case 3: 38,362, 31,651, 41,822, 41,466) Number of pixels related to curvature domain (Case 1: 30,215; Case 2: 18,209; Case 3: 24,351) | | | |

components of a shape information processing architecture. When overlaid on outlines of orientation domains, curvature domains appear distinct from orientation domains, and have a semi-regular distribution within the orientation band (*Figure 2F*, color: iso-orientation contours, gray patches: curvature domains). Similar modular curvature vs. straight preference maps were obtained in the other two cases (*Figure 2—figure supplement 4*: Case 2; *Figure 2—figure supplement 6*: Case 3).

Note that these curved grating stimuli were generated by periodically repeating an elliptical contour (see *Figure 1*, Materials and methods), resulting in, for high curvature gratings, high spatial frequency content in the flanks of the stimulus. If the curvature response was primarily due to high spatial frequency, then the curvature map and the high spatial frequency preference should have high overlap. However, this was not observed (*Figure 2H*), suggesting it is unlikely that the curvature response is primarily due to high spatial frequency content. We also examined the difference between high curvature degree and low curvature degree gratings, which have different spatial frequency content; as these maps are quite similar (*Figure 2—figure supplement 4*), it is unlikely that curvature domain response is due to spatial frequency content. We also observed that maps were stable over time, indicating these maps are unlikely to be artifactual (*Figure 2—figure supplement 5*). Furthermore, examination of images obtained in response to scrambled curved gratings minus straight gratings did not result in structured maps (*Figure 2—figure supplement 6*), suggesting that, when compared with straight gratings, it is not the presence of multiple orientations or multiple spatial frequencies alone that produced structured maps. In addition, the sizes of curvature domains (mean = 434 µm for all three cases) fall within the 200–500 µm range of functional domain sizes in V4 (*Tanigawa et al., 2010*; *Li et al., 2013*; *Ghose and Ts'o, 2017*). These data thus suggest that curvature maps are not artifactual and are distinct from previously described functional maps in V4.

We further examined selectivity of response of curvature domains by comparing magnitude of reflectance change in response to curved vs. straight grating stimuli. Reflectance change timecourses were typical of cortical intrinsic signals in V4, characterized by 2–3 s peak times and 0.01–0.03% amplitudes. As expected (*Figure 2L*, Col), color domains exhibited robust response to isoluminant color gratings (black) but weak response to achromatic gratings (gray); and failed to distinguished curved (red) vs straight (gray) gratings (Wilcoxon test, p=0.06, all three cases). Similarly (*Figure 2L*,

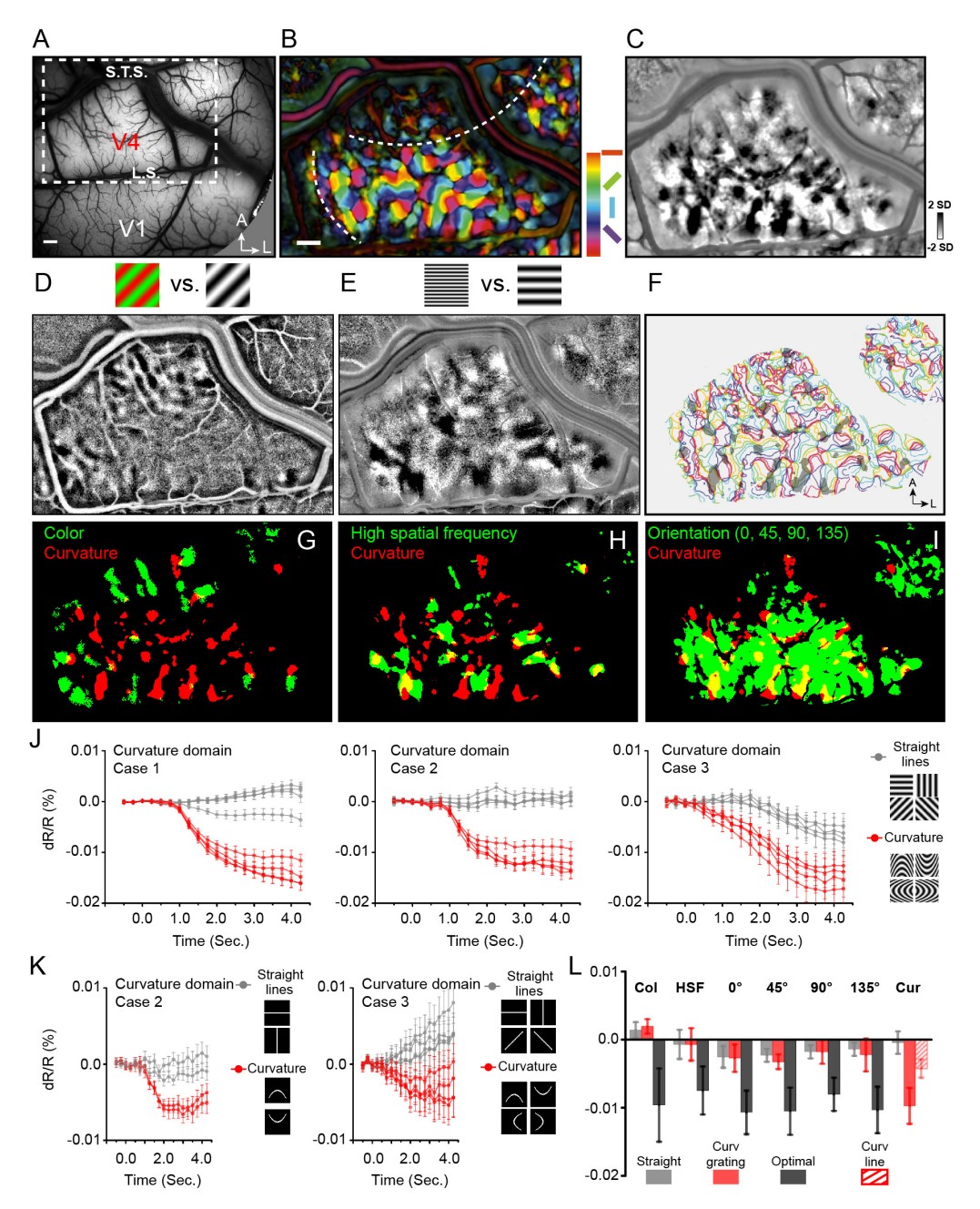

**Figure 2.** Curvature domains exist and are distinct. (A) View of cortical surface in Case 1. Dotted box: region shown in B-I. L.S., lunate sulcus. S.T.S., superior temporal sulcus, A, anterior, L, lateral. B-I: Case 1. (B) Color-coded orientation preference map. White dashed lines: approximate borders between color and orientation bands. (C) Curvature map: all curved minus all straight gratings. (D) Color preference map. (E) High spatial frequency preference map (4 cycle/deg vs. 0.5 cycle/deg). (F) Curvature domains (gray patches, two-tailed t-test, p<0.01) superimposed on iso-orientation contours. (G-I) Overlay of curvature domains (red) and G: color domains (green, from D), H: high spatial frequency domains (green, from E), I: all orientation domains (green). (J) Response time courses of curvature domains from Case 1 (left), Case 2 (middle), and Case 3 (right). Red lines: preferred stimuli. Gray lines: non-preferred stimuli. (K) Response time courses of curvature domains to flashed curved lines. Red timecourses: flashed curved lines. Gray timecourses: flashed straight lines. (L) Summary of response amplitudes for color (Col), high spatial frequency (HSF), orientation (0˚, 45˚, 90˚, 135˚), and curvature (Cur) domains shown in J, K. Gray: straight grating. Red: Curved grating. Black: optimal stimulus responses (except for Cur). For Cur, optimal response was to curved gratings (red) and to flashed curved lines (hatched red). Scale bar: 1 mm. Error bars: SEM (timecourses in J, K), SD (histogram in L).

The online version of this article includes the following figure supplement(s) for figure 2:

**Figure supplement 1.** Large field of view of functional maps in Cases 1–3.

*Figure 2 continued on next page*

*Figure 2 continued*

**Figure supplement 2.** V1/V2 and V4 represent different retinotopic locations in the same field of view.
**Figure supplement 3.** Relationship between curvature domain and low SF domain.
**Figure supplement 4.** Low and high curvature maps are highly overlapped (two Cases across two sessions).
**Figure supplement 5.** Stability of curvature response across different days.
**Figure supplement 6.** Imaging of curvature responses and scrambled map in the same cortical region (from Case 3).
**Figure supplement 7.** Supporting graphs for *Figure 2L*.
**Figure supplement 8.** Straight orientation domains and high spatial frequency domains exhibit weak responses to single curved lines.
**Figure supplement 9.** Neurons in curvature domains prefer curvature.
**Figure supplement 10.** Curvature domains were not observed in V1.
**Figure supplement 11.** Very little V2 is available on surface.
**Figure supplement 12.** Curvature domains are not pinwheel centers.
**Figure supplement 13.** Curvature domains can not be fully explained by end-stopping.

HSF), high spatial frequency preference domains responded strongly to achromatic gratings of high spatial frequency (black) but poorly to low spatial frequency gratings (gray); and failed to distinguish curved (red) vs. straight (gray) gratings (high spatial frequency domains, p=0.77, all three cases) (see *Figure 2—figure supplement 7* for supporting graphs). These analyses indicate that curvature processing is not subserved by color or high spatial frequency domains.

We also found that preferences of orientation domains and curvature domains were distinct. As expected (*Figure 2L*, 0˚, 45˚, 90˚, 135˚), *straight orientation domains* exhibited strongest response to gratings of their respective optimal orientations (black), weak responses to other straight orientations (gray), and weak response to curved gratings (red). *Curvature domains*, in contrast, exhibited strong preference for curvature stimuli. As shown in *Figure 2J* left, timecourses of curvature domains revealed strongest amplitudes for curvature gratings (red lines, Case 1), but relatively weak response to straight gratings (gray lines). Similar preferences are evident for both Case 2 (middle) and Case 3 (right). This is quantitatively summarized in (*Figure 2L*, Cur, red).

To further test whether this differential response is due to curvature, in two cases, we examined response to flashed curved and straight lines. As shown in *Figure 2K*, using the same pixels from which significant curved grating responses were obtained, we found that response of curvature domains to single flashed curved lines (red lines) was significantly greater than that to single straight lines (gray lines). Thus, curvature domains displayed preference for curved over straight contours for both grating and line stimuli, supporting a curvature-specific response. This curved line response is quantitatively summarized in (*Figure 2L*, Cur, red hatch). [Note that, in comparison, straight orientation domains and high spatial frequency domains exhibited weak responses to flashed curved lines (*Figure 2—figure supplement 8*)].

To summarize (*Figure 2L*), each of the color domains (Col), high spatial frequency domains (HSF), and orientation domains (0˚, 45˚, 90˚, 135˚) exhibit preference for their respective optimal stimulus (black bars), one which far exceeds response to curved gratings (red bars) (black vs red bars, Wilcoxon test, p<0.0001). Likewise, curvature domains (Cur) exhibit strong preference for curved gratings (large red bar) and curved lines (hatched red bar) and little response to straight gratings (gray bar). The degree of preference for curvature (Cur red vs gray) parallels the degree of preference of color, high spatial frequency, and orientation domains in V4 for their respective optimal stimulus (black vs gray, black vs red). These data show that (1) curvature domains are strongly selective for curvature stimuli and (2) non-curvature domains exhibit minimal response to curvature stimuli. Thus, curvature domains in V4 exhibit response preferences that are selective for curvature (compared to straight, color, high spatial frequency) and are maintained across two types of curvature stimulus (gratings and lines). Evidence from electrophysiological recording also show the predominance of curvature preference neurons in these domains (*Figure 2—figure supplement 9*). These curvature preferring domains were not observed in V1 (one case shown in *Figure 2—figure supplement 10*), consistent with previous studies (*Yue et al., 2014*; *Ponce et al., 2017*). As V2 in these cases was not available on the operculum (*Figure 2—figure supplement 11*), we have no data on V2.

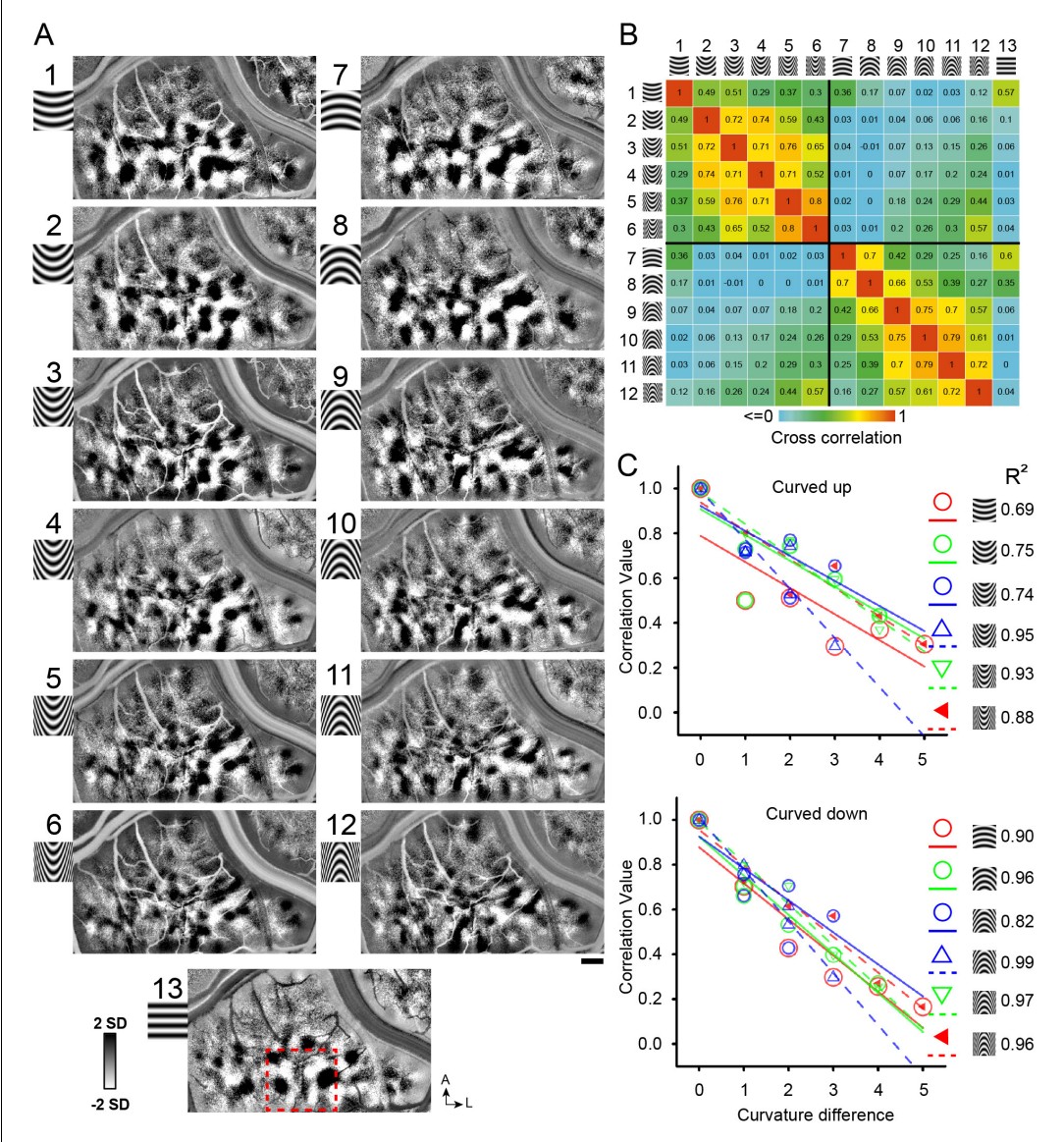

**Figure 3.** Systematic maps of curvature degree. (**A**) Maps of different curvature degrees. Each map is one curvature minus average of all straight gratings (Case 1). 1–6 and 7–12: upwards and downwards curvatures, respectively, from low to high curvature preference, 13: straight grating. Red dotted square marks the region that is further analyzed in *Figure 7A,B*. Correlation values for pairs of curvature response maps (from A). Color bar: high (red) to low (blue) correlation values. (**C**) The more similar the curvature the greater the correlation value. X axis: curvature degree difference. Y axis: correlation value. Color symbols: correlation value for each curvature above with respect to its curvature degree distances (each fit with matching color line; regression values are shown on the right). The fitted lines are not meant to indicate linear fits; rather, they help to see the trends. Scale bar, 1 mm.

The online version of this article includes the following figure supplement(s) for figure 3:

**Figure supplement 1.** Correlation values between pairs of straight orientation maps obtained in three different runs (from Case 1).

**Figure supplement 2.** Systematic maps of curvature degree and curvature orientation in other cases.

## Mapping types of curvature

We predicted the global organization of curvature representation should systematically shift with changing curvature degree and with changing curvature orientation. To address this global question, we first tested whether the response pattern would vary with changes of curvature features. We conducted cross-correlation of images obtained with different curvature degrees and with different curvature orientations. By examining correlations between images, we expected that increasing

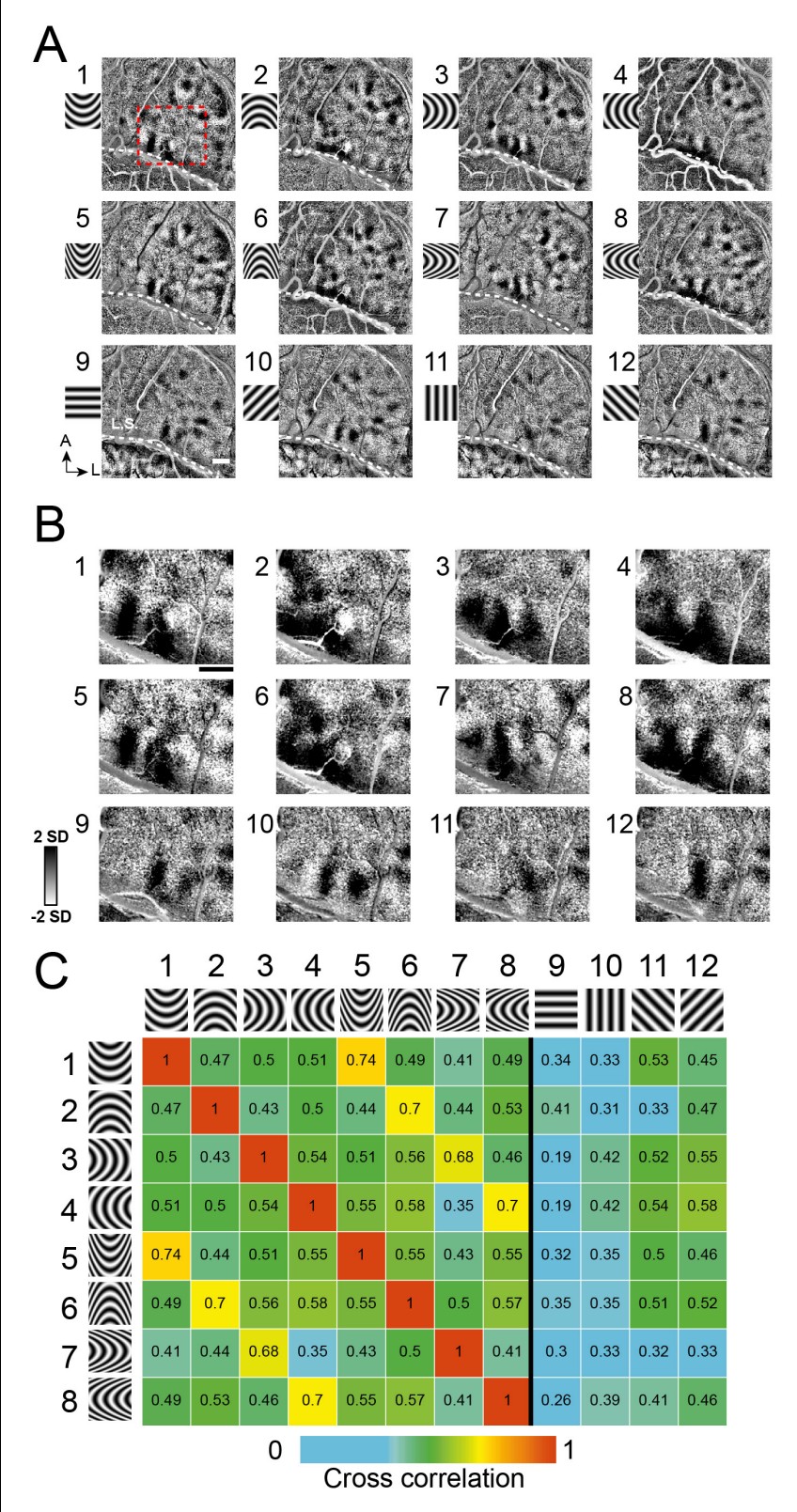

**Figure 4.** Systematic maps of curvature orientation. (**A**) Imaging results of the responses to different curvature orientations. Maps of curvature minus average of straight gratings (Case 2). 1–4: low curvature degree, a/b ratio = 2, 5–8: high curvature degree, a/b ratio = 5. 1,5: upwards. 2,6: downwards. 3,7: leftwards. 4,8: rightwards. 9–12: straight. (**B**) Enlarged view of the cortical region outlined by red dotted box in **A**. (**C**) Correlation values for

*Figure 4 continued on next page*

*Figure 4 continued*

pairs of maps (from A). Maps of similar curvature orientation have high correlation values, while those with different curvature orientations have low correlation values. Colors code correlation values (high, red, to low, blue, see color bar).

stimulus similarity should predict increasing map similarity. In this section, we show that this prediction is met. In the next section, we zoom in on a small location region to study the local organization more closely.

## Low to high curvature maps

To examine whether there could be a change of curvature domains representing low to high curvatures, we designed a series of curved gratings from low to high curvature (*Figure 3A*, upwards: 1–6; downwards: 7–12, Case 1; *Ponce et al., 2017*) and presented these stimuli in two cases. Maps obtained in response to these curvature stimuli exhibited regular structure and domain size consistent with functional maps (see *Figure 3A*). To examine the possibility of a low to high curvature gradient, we conducted cross correlations between pairs of these images (see Materials and methods). We reasoned that, if a gradient of curvature degree exists, then there should be a gradually

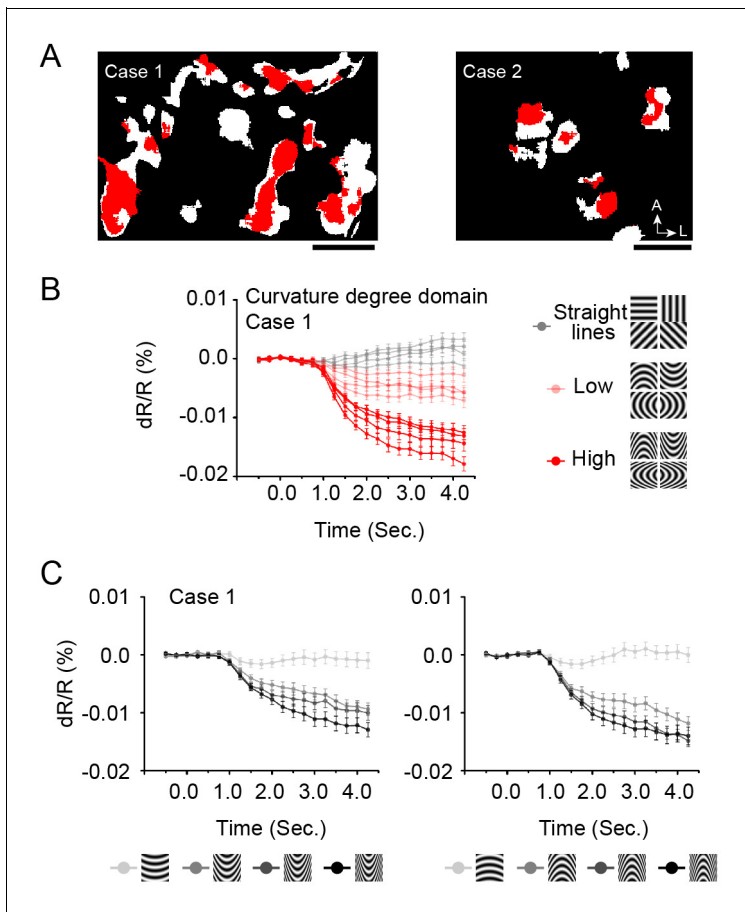

**Figure 5.** Subregions of high curvature preference. (**A**) Curvature preferring pixels. Red pixels: high curvature preferring pixels (high curvature vs. Low curvature, a/b ratio = 5 vs. 2, all four orientations, two-tailed t-test, p<0.01). White pixels: (curvature vs. straight). (Case 1: left, Case 2: right, two-tailed t-test, p<0.01). (**B**) High curvature subdomains (red pixels in **A**, Case 1) prefer high curvature (dark red lines, four orientations, a/b ratio = 5) over low curvature (light red lines, four orientations, a/b ratio = 2), and straight lines (Gray lines, four orientations). (**C**) Timecourses of response to 4 degrees of curvature (from high to low, darkest to lightest gray). Scale bar, 1 mm. Error bar: SEM.

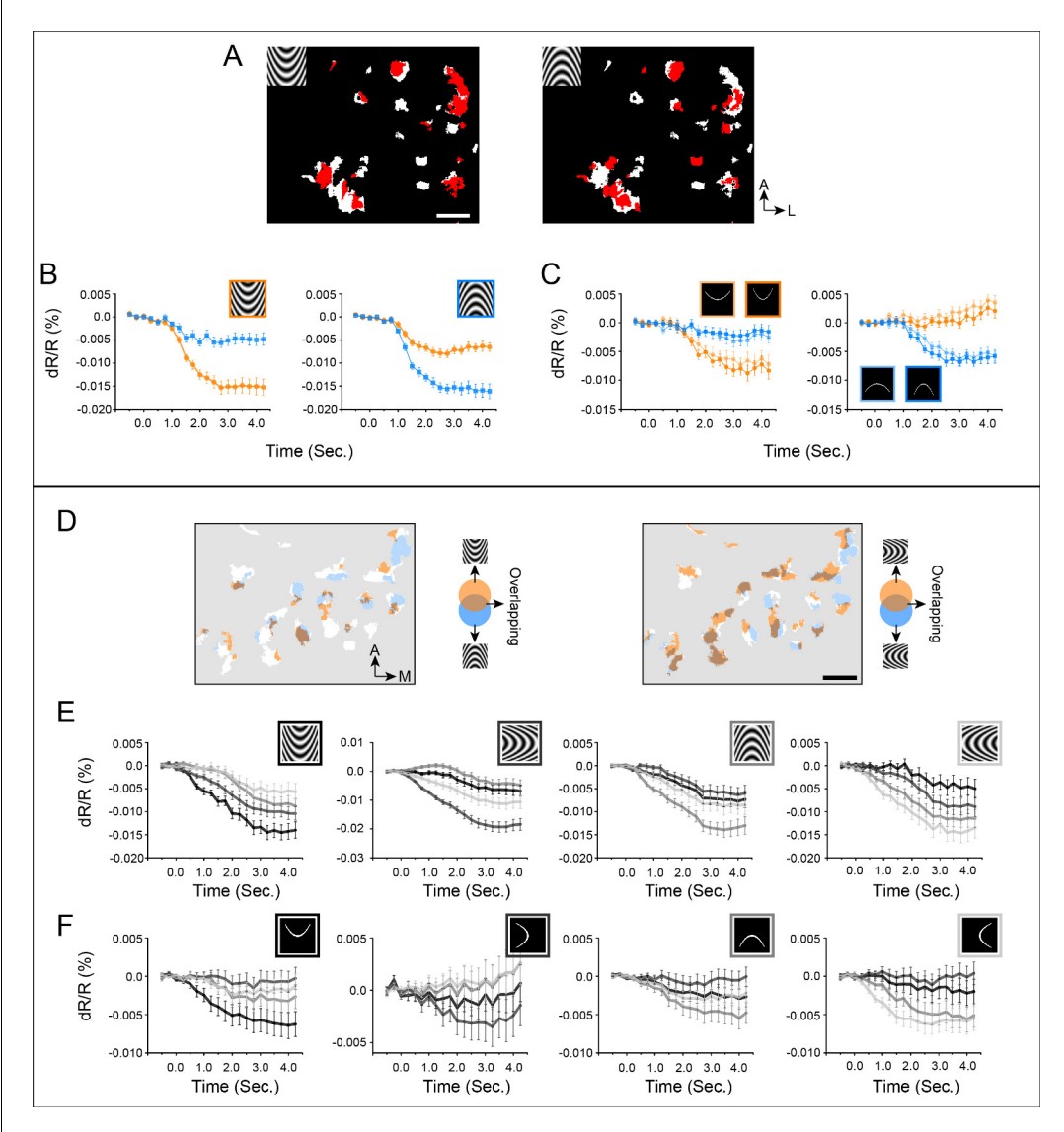

**Figure 6.** Organization of curvature orientation representation in V4 curvature domains (**A-C**: Case 2, **D-F**: Case 3). (**A**) Curvature orientation maps (Case 2). White pixels: curvature vs. straight (two-tailed t test, p<0.01). Red pixels: significantly activated by different curvature orientations (compared to average of four straight orientations, two tailed t-test, p<0.01). There are overlapping as well as non-overlapping pixels between two opposite orientations. (**B**) Response timecourses of red pixels in A to curved gratings. (**C**) Response timecourses of the same red pixels in A to curved single lines. Orange lines: responses to up curvature, Blue lines: responses to down curvatures. For flashed curved lines, we used two different curvature degrees. (**D**) Curvature orientation maps (Case 3). White pixels: curvature vs. straight (two-tailed t test, p<0.01). Colored pixels: significantly activated by different curvature orientations. (**E**) Response timecourses of colored pixels in maps above to curved gratings. (**F**) Response timecourses of colored pixels in maps above to single flashed curved lines. Black lines: response timecourses to upwards curvature. Dark gray lines: response timecourses to leftwards curvature. Gray lines: response timecourses to downwards curvature. Light gray lines: response timecourses to rightwards curvature. Comparison of E and F graphs show that the orientation of the best curved grating response matches that of the best curved line response (gray timecourses). Scale bar: 1 mm. Error bar: SEM.

changing map similarity as curvature degree changes. Correlation values ranged from 0 (no correlation or minor negative correlation, blue) to 1 (two maps are identical, red) (see color bar). To give the reader an idea of what high map correlations values are, the range of correlation values for four straight orientation maps in V4 shown in *Figure 3—figure supplement 1*; matched orientation correlations range from 0.66 to 0.9 (yellow, orange, red values). As shown in *Figure 3B*, higher correlation values tended to occur between curvature maps of similar curvature degree (orange and yellow squares in upper left quadrant and lower right quadrant) and lower correlation values between those

of dissimilar curvature degree (green and blue squares in upper left quadrant and lower right quadrant). Within this correlation matrix, the lowest correlation values were obtained between opposing curvatures (upper right quadrant and lower left quadrant) and between curved and straight stimuli (rightmost column). High correlation values were obtained between adjacent 1 or two curvatures (orange and yellow boxes) (e.g compare values between pairs 2–3, 3–4); correlation values dropped off with increasing differences in degree of curvature (green and blue boxes) (e.g. 2–3, 2–4, 2–5). This relationship is plotted in *Figure 3C* for each of the 6 Curved Up (top graph) and 6 Curved Down (bottom graph) curvature degrees (e.g. *Figure 3C* top graph red circles: show declining correlation with curvature difference). This declining correlation with curvature difference suggests the presence of a shifting map of curvature degree. Similar results were obtained in a second case (Case 2, *Figure 3—figure supplement 2A–B*).

## Curvature orientation maps

In another experiment (*Figure 4A,B*, Case 2), we used four *orientations of curvature* (curved upwards, downwards, leftwards, and rightwards), each at low curvature (1-4) and high curvature (5-8). Similar to maps for straight orientation (*Figure 3—figure supplement 1*), we expected stronger correlation between maps of similar curvature orientation and lower correlation for dissimilar orientations. This expectation was supported by computed correlation indices for pairs of images (*Figure 4C*). These correlation values illustrate that high correlation values (orange, yellow) occur only between curvature gratings of the same orientation (e.g. 1–5, 2–6), and low correlation values occur between curvatures of different orientations (green boxes in columns 1–8, e.g. 1–2, 3–4). The finding that the low correlation values occur between these curved and straight gratings (columns 9–12) supports the distinctness of straight vs. curved contour representation. Similar results were obtained in a second case (Case 1, *Figure 3—figure supplement 2C–D*).

## Mapping central part of curvature grating

To more precisely study the organization within curvature domains, we then focused on the cortical region of V4 representing the center of the curvature stimuli (the region of high curvature content, ~1 deg). As shown in *Figure 2—figure supplement 2*, using thin vertical and horizontal lines, we determined the retinotopic representation of V4 within the imaged field of view. The curvature stimulus was then centered precisely on this location. Note that the region of V1 visible in the same field of view represents a shifted topographic location, so we were unable to map response to the curvature stimulus in both V1 and V4 simultaneously. In these cases, V2 was buried in the lunate sulcus and was not visible.

## Organizations within curvature domains

By focusing on the region representing the central part of the curvature stimulus, we were able to discern subdomains of high curvature vs. low curvature preference and largely non-overlapping subdomains of opposing curvature orientation. *Figure 5* shows that curvature domains (*Figure 5A*, Case one and Case 2, white pixels: curvature vs. straight) contain subregions of high curvature preference (red pixels: high curvature vs. low curvature). This is shown by preference (*Figure 5B*, Case 1) for high (dark red lines) over low (light red lines) curvature and even weaker response to straight gratings (gray lines). Moreover, the response magnitudes of these subregions gradually diminish with curvature degree difference (*Figure 5C*, dark to light lines plot responses of high to low curvature; left: upwards curvature, right: downwards curvature). Thus, there are high curvature subdomains within curvature domains (*Figure 5A*), and curvature subdomains exhibit graded response to curvature degree (*Figure 5B,C*).

   *Figure 6* illustrates the finding that paired opposing curvature orientations occupy roughly complementary positions within the curvature domains. An example of an UP/DOWN pair is shown (*Figure 6A*). *Figure 6B,C* show that pixels that prefer upwards maintain their preference across curved gratings (6B) and two degrees of curved lines (6C). This is also true for downwards pixels (6B, C). A similar result is shown in *Figure 6D–F* (Case 3). For four curvature orientations, upwards/downwards (*Figure 6D* left panel) and leftwards/rightwards (*Figure 6D* right panel) pairs occupy roughly complementary regions within the curvature domains. These preferences are maintained across curved grating and curved line stimuli. Additionally, the graded response for each of the four

orientations suggests the presence of a curvature orientation organization (see below). These results reinforce the presence and selectivity of curvature orientation domains. We note that this complementarity is not absolute. Within the curvature domains, the average percentage of pixels responsive to two opposite orientations (overlapped pixels vs. all colored pixels) was 22.4% (Case 1: 32.1%; Case 2: 14.9%; Case 3: 20.3%).

## Organization of curvature representation

Given that there are indications of shifting maps for curvature degree, we examined whether we could discern any systematic relationship between domains for straight orientation and curvature degree. We hypothesized that in some parts of V4, there is a gradual shift from straight to curved representation. In another experimental session, as shown in *Figure 7A* (Case 1), within the central region of curvature representation, we obtained (at a single orientation) curvature minus straight orientation maps to different degrees of curvature. Within this region, there appear to be several progressions of straight to curved domains (four straight-to-UP and four straight-to-DOWN). Overlays of these maps are summarized in *Figure 7B* (left panel, top row: UP overlays, bottom row: DOWN overlays; white regions: curvature domains). A subset of these overlays are shown in *Figure 7B* (middle and right panels. UP: top row, DOWN: bottom row). The middle panels isolate straight orientation domains (blue) and the nearest curvature domains (from straight to high curvature, color code at top) (domain clusters are separated for clarity). The right panels demarcate the center of each domain with a color-coded dot (center of mass). We observed a spatial progression from the center of the straight orientation domain (blue dot in center of blue dotted line) to low curvature domain (blue-oranges) to high curvature domain (orange); these colored dots are connected by a line (shaded from blue to orange) to illustrate the progression. Four examples of such progressions for Up are shown in the top panel and four progressions for Down in the bottom panel. Progressions arising from a single orientation domain can lead to two different curvature domains (e.g. UP: leftmost orientation domain, Down: rightmost orientation domain). Additional examples are shown in *Figure 7—figure supplement 1*. These progressions suggest a possible systematic contour representation in V4.

To further examine the details of these progressions, we analyzed the distances between each straight orientation domain and the nearest (same orientation) curvature domains. Specifically, starting from a straight orientation domain (n = 8), we selected the nearest domain of each curvature condition (shown in *Figure 7B*) and measured the distances between center of mass of each domain and that of the straight domain (leftmost panel in *Figure 7C*). These values are plotted in *Figure 7D* (gray lines). This reveals that the curvature preference domains generally progress from low to high curvature with distance across the cortex. Note that some parts of these curvature sequences shift and other parts do not appear to shift. However, on average (black line), across these eight straight-to-curved sequences, there is a general trend of greater distance with increasing curvature. We conducted this analysis for the other two cases (shown in *Figure 7—figure supplement 1*). Case 2 shows three progressions with a similar tendency to shift (*Figure 7—figure supplement 1B*) and one without shifts (*Figure 7—figure supplement 1C*) (graphed in *Figure 7—figure supplement 1D*). Case 3 shows seven progressions with shifts (*Figure 7—figure supplement 1F*), and one without (*Figure 7—figure supplement 1G*) (graphed in *Figure 7—figure supplement 1H*). In total across the three cases, on average, there is a general overall tendency for straight-to-curved domains to exhibit spatial shifts. However, there are some domains with broad curvature preference. This suggests the presence of a diversity in organization, indicating that curvature representation in V4 is complex.

## Curvature domains are not orientation pinwheel centers or sums of orientation components

We considered the possibility that curvature responses could be due to activation of a mixture of oriented responses, such as that found at orientation pinwheel centers. As shown in two cases in *Figure 2—figure supplement 12* (Case 1: A-B, Case 2: C-D), the curvature domains (orange regions) do not co-localize with orientation pinwheel centers (blue dots). In fact, most of the pinwheel centers are outside the curvature domains entirely. Moreover, as shown by the responses of pinwheel centers to straight and curved stimuli (*Figure 2—figure supplement 12E*), the pinwheel center

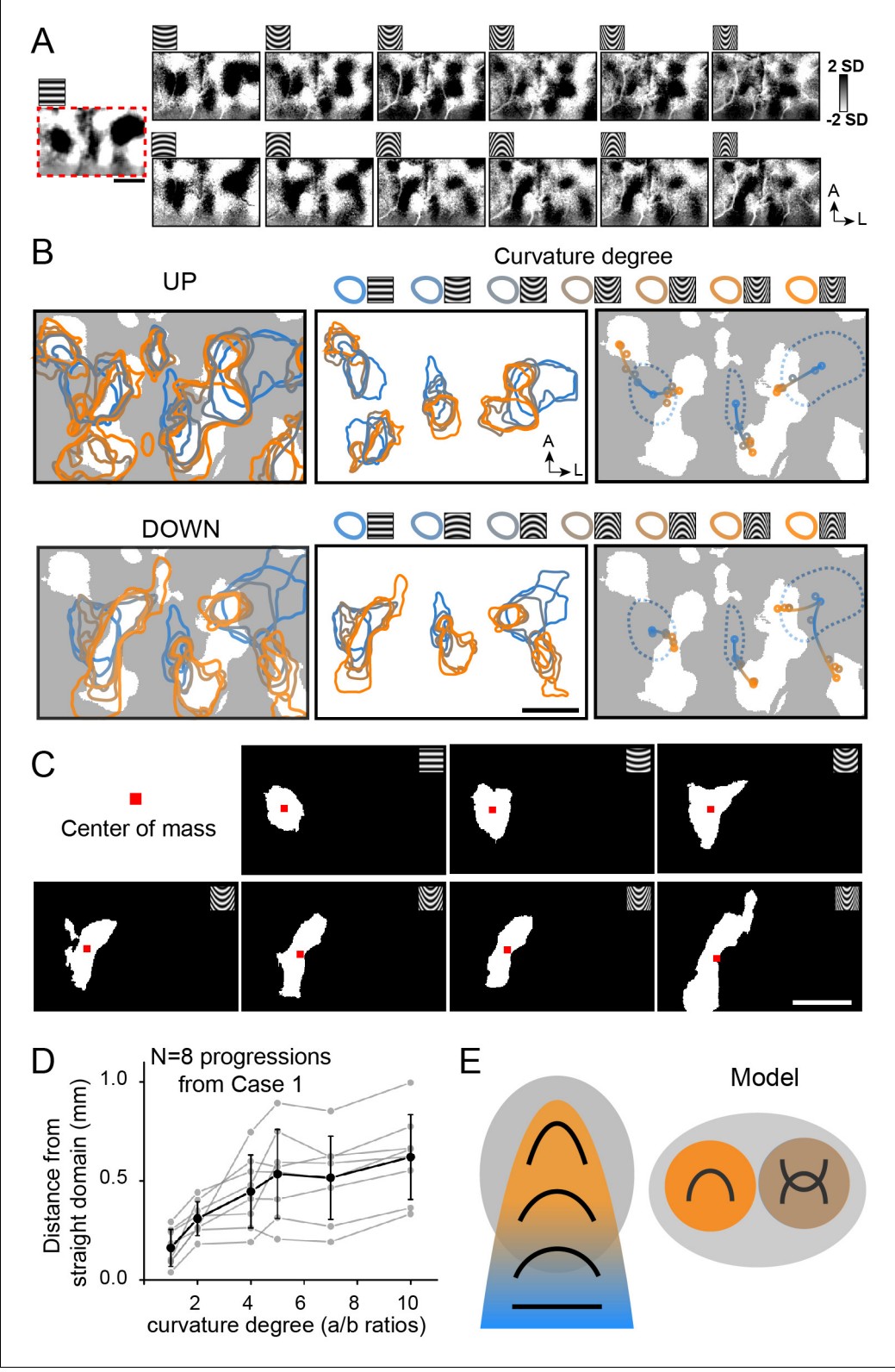

**Figure 7.** Functional organization of curvature in V4. (**A**) Maps of different stimuli minus average of straight gratings (Case 1). Top row: upwards curvatures, bottom row: downwards curvatures. Leftmost: horizontal gratings. (**B**) Maps of progressions from straight to curved representation. Top row: responses to upwards curvatures (Up). Bottom row: responses to downwards curvatures (Down). *Left panels:* activated regions (two-tailed t-test, p<0.01) *Figure 7 continued on next page*

*Figure 7 continued*

corresponding to different curvature stimuli. White regions: curvature domains. Color code: high (orange) to low (blue). *Middle panels:* activated regions corresponding (two-tailed t test, p<0.01, curved vs. average of straight) to respective curvature degrees are outlined by different colors (domain clusters are separated for clarity). Color code (at top): high (orange) to low (blue). [Note that the two leftmost domain progressions in Up panel are associated with the same orientation domain but are separated for clarity.] *Right panels:* Location of the activation center of each domain (indicated by colored dot). White regions: curvature domains. Blue dotted lines: horizontal orientation domains. Shifting progressions from straight orientation (blue dot) to low curvature (blue-orange dot) to high curvature (orange dot) are observed, as indicated by colored line (shaded from blue to orange). (C) Regions that were significantly activated (two-tailed t test, p<0.01) by corresponding stimuli (from Case 1). Red dots: centers of mass. (D) We measured the distance between each curvature domain activation center of mass and its corresponding straight orientation domain center of mass (a/b rations: 1, 2, 4, 5, 7, 10 in Case 1, eight progressions). Gray dots and lines: results from different progressions, black dots: averaged across all eight progressions. Error bar, SD. (E) Summary of curvature domain findings. Left: curvature degree progressions, Right: presence of curvature domains and complex curvature domains. Gray oval: curvature domains.

The online version of this article includes the following figure supplement(s) for figure 7:

**Figure supplement 1.** Diversity of curvature domain organization (A-D: Case 2, E-H: Case 3).

**Figure supplement 2.** Curvature domain response is not due to component orientation response.

responses to curvatures are weak and do not distinguish curvature from straight (Wilcoxon test, p=0.13). This makes it unlikely for pinwheel centers to be locations of curvature response. We also considered the possibility that curvature responses are simply a weighted sum of responses to component orientations. This is unlikely because (1) the straight orientation domains are spatially distinct from the curvature domains (*Figure 2*). And (2) our analysis of response to component orientations compared with response to curvature shows that curvature domain response to straight component orientations is weak (smaller by roughly an order of magnitude) (*Figure 7—figure supplement 2*).

## Curvature domains are not end-stopping domains

There is strong support for the role of end-stopped cells in curvature representation (*Hubel and Wiesel, 1965*; *Hubel and Livingstone, 1987*; *Dobbins et al., 1987*; *Sceniak et al., 2001*). The rationale is that straight contours extend into the RF end zones leading to suppression, while curved contours do not, making end-stopped neurons ideal 'not straight' candidates for encoding curvature. Electrophysiological study of single unit responses to curvature gratings in V4 finds strong correlation between curvature preference and strength of end-stopping (*Ponce et al., 2017*). Thus, this study suggests response to these two parameters in V4 are indistinguishable, raising the possibility that the curvature domains recorded here actually represent end-stopping response.

We feel this is unlikely. End-stopping, while an essential contributer to curvature computation, is not sufficient for all curvature encoding. As supported by previous studies in V4 (*Nandy et al., 2013*; *Nandy et al., 2016*), the curvature selectivity displayed by many V4 neurons can also be caused by other properties (e.g. complex structures of receptive field). While end-stopped cells may contribute to the curvature domain responses recorded here, curvature domains *are not equal to* end-stopping domains. As shown in *Figure 2—figure supplement 13*, comparison of end-stopping maps (1 deg vs 4 deg straight gratings) and curvature maps reveal that they are distinct. *Figure 2—figure supplement 13C* (overlay of A, end-stopping map and B, curvature map) illustrates that there are some regions of overlap (~33.3%), but 66.7% of domains are in spatially distinct locations. Moreover, quantification of curvature-preferring pixels (*Figure 2—figure supplement 13D*) reveals that, while there is a preference for the small over the large straight stimulus (compare two light gray lines), the responses to straight stimuli are still weaker than that to curved (black lines). This suggests that, in V4, curvature domains are not simply end-stopped domains.

# Discussion

## Summary

In summary, we provide evidence for the existence of curvature domains in V4 and show that there is a systematic map of curvature representation in V4. To the organization of curvature

representation within V4, we designed stimuli that spanned curvature degree and curvature orientations. Instead of using shapes with complex structures such as hyperbolic/polar gratings (*Hegdé and Van Essen, 2003*) or forms composed of multiple corners and lines (*Pasupathy and Connor, 2002*; *Brincat and Connor, 2004*), these were stimuli with relatively simple curvature content (similar to *Ponce et al., 2017*). The gradations of curvature degree across our stimulus set revealed that response preference is spatially mapped in certain parts of V4 (from straight to highly curved, *Figure 7B* and *Figure 7—figure supplement 1B,F*). We showed that maps for curvature are distinct from those for straight orientation, color, high spatial frequency, orientation pinwheels, and end-stopping. These maps also have the spatial appearance of organized functional maps in terms of domain size and distribution. We find curvature domains exhibit selectivity for degree of curvature (low to high), one which maps in a spatially shifting manner in V4. These progressions are not necessarily linear, as indicated by the distance changes related to curvature degree (*Figure 7D*, *Figure 7—figure supplement 1D,H*). In addition to these curvature degree progressions, other types of spatial organizations were also observed (*Figure 7—figure supplement 1C,G*). Similarity of response preference across both curved gratings and single curved lines provide further evidence for curvature selectivity rather than some other aspect of these stimuli.

## Hypercolumn model

While the overall spatial relationship between straight and curved features will take additional study to fully establish, given the data at hand, we propose the model shown in *Figure 7E*. We suggest that straight (large gray square) and curved (large gray oval) regions in V4 co-exist and occupy roughly complementary territory within the 'orientation bands' of V4. Within the curvature domains, there are multiple types of organizations, characterized by sequences of curvature degree as well as domains with complex curvature preference. To our knowledge, no previous study has provided evidence suggesting that curvature is a basis for a hypercolumn in V4. This framework will need to be further tested in future studies.

## Testing other possible interpretations
### Unlikely to be primarily component orientation responses
We tested the possibilities that the curvature preference domains are actually responses to the weighted sum of component straight orientations. However, we found little support for this: the weighted sum of three primary components produced responses that were an order or magnitude weaker than to the curved stimulus (*Figure 7—figure supplement 2*). Scrambled stimuli which contain straight components failed to produce structured maps (*Figure 2—figure supplement 6*). And orientation pinwheels, which contain many straight component responses, were at locations quite distinct from the curvature domains (*Figure 2—figure supplement 12*). Neither did our data support the possibility that curvature domains are end-stopping domains (*Ghose and Ts'o, 1997*), as these mapped to spatially distinct locations (*Figure 2—figure supplement 13*). Thus, it does not appear that curvature domain responses are due to a simple linear combination of local orientation domain responses, but result from a non-linear (perhaps multi-step) integration of straight oriented inputs.

### Unlikely to be high spatial frequency domains
We consider the possibility that the curvature responses are actually responses to the high spatial frequency aspects of the curvature gratings. Based on curvature stimuli used in *Ponce et al., 2017*, we designed curvature gratings composed of ellipsoid curves repeated sinusoidally. As shown in *Figure 1*, different stimuli with different degrees of curvature created in this fashion have common spatial frequency along the center axis. Towards the flanks of the stimulus, especially for higher curvature, the grating has higher SF content than that on the central axis. However, several observations make our results inconsistent with the hypothesis that these results are due primarily to these high SF aspects of the stimuli. (1) High spatial frequency maps in V4 (first reported by *Lu et al., 2018*) differed from curvature maps (*Figure 1H and L*). On average, less than 15% overall from three cases exhibited overlap between high spatial frequency domains and curvature domains. (2) Similar maps were obtained to gratings with different degrees of curvature (*Figure 2—figure supplement 4*). If the results were due primarily to spatial frequency content, the maps resulting from high and

low curvature gratings should differ. (3) Similar maps were obtained to gratings and single lines of the same curvature (*Figure 2K,L*, *Figure 6*), although gratings and single lines have different spatial frequency contents. (4) Maps obtained in response to scrambled curvature stimuli (which also contain high spatial frequencies but lack the structural curvature information) failed to produce curvature maps. Thus, while we have not excluded all possibilities, we believe that the bulk of the evidence provides a consistent view. Note also that these curvature domains have little overlap with low spatial frequency domains (*Figure 2—figure supplement 3*), further indicating that a representation of curvature and spatial frequency content are distinct.

## Unlikely to be end-stopping domains

As previous studies suggested, size tuning or end stopping may play an important role in curvature detection (*Hubel and Wiesel, 1965*; *Dobbins et al., 1987*) and is not independent from curvature preference features (*Ponce et al., 2017*). However, other studies have suggested neurons with complex receptive field structures may also account for curvature preferences (*Nandy et al., 2013*; *Nandy et al., 2016*). For this reason, while end-stopping likely contributes to curvature response, there are also other relevant parameters contributing to curvature response. (1) Surround suppressed and curvature domains are not the same: Similar to previous studies (*Ghose and Ts'o, 1997*), we found there were size sensitive regions in V4 (Case 2, *Figure 2—figure supplement 13A*). However, while there was some overlap, the size sensitive regions were on the whole spatially distinct from the curvature domains (compare *Figure 2—figure supplement 13A and B*, overlay in C). (2) Weak response to small stimuli. Responses of curvature domains to small stimuli were weak in comparison to response to curvature stimuli (*Figure 2—figure supplement 13D*), suggesting weak end-stopping. Therefore curvature response can not be fully explained by strong responses to end-stopping. (3) In addition, if end-stopping was the only factor that matters, curvatures with opposite orientations would densely overlap, which is not the case (*Figure 6*). Thus curvature preference response in V4 is not due to end-stopping alone.

## Curvature domains are signature modules of V4

Our results are bolstered by another recent study which also reports the presence of curvature domains in V4 (*Tang et al., 2020*). Using stimuli that included simple shapes (e.g. circles vs triangles) and shape components, the authors also report that curvature domains are distinct from orientation domains and color domains, and are absent in V1 and V2. Using two-photon imaging, the authors found neurons in curvature domains respond to a diversity of curved shapes and curve parts, and exhibit weak response to straight lines and corners. Within a curvature domain, they find subclusters of neurons with similar shape or curve part preference. Interestingly, some clusters exhibit preference for shapes (circles) over shape parts (curved portions of circles), raising the possibility that some domains in V4 may represent simple shapes. While the authors did not systematically explore the organization of curvature degree or curvature orientation, this elegant study further reinforces the concept that curvature domains are a primary feature of V4 organization.

## Significance of modules—cortical modules represent key transformational stages in the visual hierarchy

We are excited that the combination of MRI (*Op de Beeck et al., 2008*; *Popivanov et al., 2015*; *Yue et al., 2014*; *Bao et al., 2020*), optical imaging (*Ghose and Ts'o, 1997*; *Tanigawa et al., 2010*; *Lu et al., 2018*; *Jiang et al., 2019*; this study; *Tang et al., 2020*), and two-photon (*Jiang et al., 2019*; *Tang et al., 2020*) scale studies are providing a functional organizational view of shape processing in the ventral visual pathway. While there is much yet unknown about curvature and shape representation, there is enough to see a sketch of the organizational hierarchy. By conducting fMRI mapping in response to a large array of stimuli containing curved (rounded, spheres, faces) vs rectilinear (square shapes, pyramid arrays, buildings) objects, *Yue et al., 2014* reported the presence of three ~ 1–2 cm-sized curvature patches in the temporal lobe of macaque. These patches revealed a hierarchy of increasing curvature complexity from the 'posterior curvature-biased patch' in near-foveal V4 (PCP), to a middle curvature-biased patch (MCP) in posterior STS, to an anterior curvature-biased patch (ACP) in anterior TE.

The region of V4 studied here corresponds well with Yue et al's PCP patch. Here, zooming in to the several mm scale, we find 200–500 µm-sized curvature degree and curvature orientation domains within (what is likely) Yue's PCP patch (see Yue *Figures 2* and *3*). Consistent with the lack of curvature response in V1, V2 (*Yue et al., 2014*; *Hegdé and Van Essen, 2007*; *Ponce et al., 2017*), we found little evidence for curvature response preference in V1. Thus, much as V1 is considered the 'orientation-emergent' area, and V2 the 'cue-invariant orientation-emergent' area, we concur with the view that V4 is the 'curvature-emergent' area. Integration of signals from V4 curvature domains are likely to contribute to additional levels of systematic maps associated with lower (MCP, PFP) and higher (ACP, AFP) order object, face, and body patch representations (*Bao et al., 2020*). We predict that these higher level patches will also contain systematic columnar maps.

## Materials and methods

Data was acquired from three hemispheres of two adult macaque monkeys (one male and one female, Macaca mulatta). All procedures were performed in accordance with the National Institutes of Health Guidelines and were approved by the Zhejiang University Institutional Animal Care and Use Committee.

### Animal preparation

Chronic optical chambers were implanted above the area V4d containing lunate sulcus, superior temporal sulcus (see *Figure 2A*) as described previously (*Li et al., 2013*). The only difference is that we used transparent glass instead of a nylon chamber. The eccentricity of the visual field corresponding to the exposed V4/V2/V1 was 0–5°. Following the craniotomy surgery, optical images were collected during which basic functional maps as well as curvature responses of V4 were obtained. Monkeys were artificially ventilated and anesthetized with propofol (induction 5–10 mg/kg, maintenance 5–10 mg/kg/hr, i.v.) and isoflurane (0.5–1.5%). Anesthetic depth was assessed continuously via monitoring heart rate, end-tidal $CO_2$, blood oximetry, and eeg. Rectal temperature was maintained at around 38C°. Animals were paralyzed (vecuronium bromide, induction 0.25 mg/kg, maintenance 0.05–0.1 mg/kg/hr, i.v.) and respirated. Pupils were dilated (atropine sulfate 1%) and eyes were fitted with contact lenses of appropriate curvature to focus on a stimulus screen 57 cm from the eyes.

### Visual stimuli for optical imaging

Visual stimuli were created using ViSaGe (Cambridge Research Systems Ltd.) and displayed on a calibrated 27-inch monitor (Philips 272G5D) running at 60 Hz refresh rate. The luminance for white stimuli was 206.52 cd/m$^2$ and black was 0.50 cd/m$^2$. Full-Screen visual grating stimuli were used to locate color preference domains in V4. Red/green isoluminance and black-white sine-wave drifting grating stimuli were presented at two different orientations (45° and 135°) with the same spatial frequency (0.5 or one cycles/°), temporal frequency (2 or 4 Hz) and mean luminance level. To acquire orientation maps and spatial frequency (SF) maps, gratings with four different orientations (0°, 45°, 90°, 135°, see *Figure 2J*) and two different SFs (0.5, 4 cycles/deg) were presented.

Curvature stimuli were designed based on the formula of ellipse, $X^2/a^2 + Y^2/b^2 = 1$, where a and b represent the length of the long axis and short axis, respectively. Different a/b ratios were used (2, 5, see *Figure 5B*; 1, 4, 7, 10, see *Figure 5C*) to create different curvature degrees. The ellipses (see *Figure 1*) were cropped and duplicated to generate a curvature grating template. SF (along the long axis) of the curvature stimuli was varied by changing the spacing of the curves. The curvature grating was drifted by moving the stimulus window along the curved grating template (see *Figure 1*). Drifting straight gratings were also created by this process and had the same luminance, drift speed, and spatial frequency (at center axis of curvature) as curvature stimuli. For curvature maps, the position of the stimuli on the monitor was first determined. As a control for curvature, scrambled versions of a curvature grating were created by dividing the grating into 64 subunits and randomly rearranging the locations of these subunits (*Figure 2—figure supplement 6A*).

#### Retinotopic mapping

As we used vecuronium bromide during the experiment, the two eyes did not look at the same position of the monitor, thus for curvature mapping, we covered one eye and only presented the

curvature stimuli to one eye. In every experiment session, we first mapped the RF location of V4 region and presented the stimuli at that location (~1–5 deg eccentricity in V4). For placement of stimuli on the monitor, we mapped the retinotopy of V4 using a series of 0.2° width horizontal or vertical dashed lines (SF = 1 cycles/°) located at different positions to determine the stimulus center (locating the stimulus center at the imaged V4 center, *Li et al., 2013*, also see *Figure 2—figure supplement 2*). As the receptive field sizes of V4 neurons at our recording position are around 3–5 degrees (*Chen et al., 2014*), our stimuli were set to 4° and presented monocularly during the experiment. Flashed curved lines were also 4° in size (for each line, the length was 4°). The center of straight lines or the vertex of curved lines were fixed at the position that we chose during the retinotopic mappings. The stimuli were presented at 4 or 8 Hz interleaved with black screen.

## Optical imaging

The brain was imaged through the implanted glass. Images of cortical reflectance changes (intrinsic hemodynamic signals) corresponding to local cortical activity were acquired (Imager 3001, Optical Imaging Inc, German town, NY) with 632 nm illumination. Image size was 1080 × 1308 pixels representing 14.4 × 17.4 (case 2) or 8.7 × 10.5 (cases 1 and 3) mm field of view (see *Figure 2—figure supplement 1*). Visual stimuli were presented in a random order. Each stimulus was presented for 3.5 (for color and high spatial frequency stimuli) or 4.5 (curvature and corresponding straight stimuli) seconds. The imaging data were stored in a block fashion. Each block contained the imaging data recorded from all the stimulus conditions (presented one time). For functional domains, each stimulus was presented at least 30 times. For retinotopic mapping, each stimulus was presented 10 times. Imaging started 0.5 s before the stimulus onset and ended till the stimulus offset with a sampling rate of 4 Hz. In V4, cortical fields of view were ~1–5 deg eccentricity; in V1, locations were at about 1–2 deg eccentricity. For all three cases, very little V2 was available on the surface and so was unavailable for imaging (see *Figure 2—figure supplement 11*).

## Map-guided cell recording

Under a surgical microscope, a tungsten microelectrode (impedance 1–4 MU at 1 kHz, FHC) was lowered into the cortex (manipulator: Narishige MO-10) targeting the center of a curvature domain. Neural activity was amplified at 1 k or 10 k gain (Model 1800, A-M Systems) and digitized at a sampling rate of 30 kHz (Blackrock microsystems). Each stimulus was normally tested for 15–30 trials.

## Data analysis

### Functional maps

With the following formula, $\Delta \mathrm{R}_i = (\bar{R}_{i1} - \bar{R}_{i2}) \times \sqrt{N}/S_i$, we assessed the response differences between two comparison groups. $\bar{R}_{i1}$ and $\bar{R}_{i2}$ are the mean dR/R values ($dR/R = \frac{R_{8-16} - R_{1-3}}{R_{1-3}}$, R$_{8-16}$ is the averaged response from frame 8 to 16, R$_{1-3}$ is the averaged response from frames 1 to 3) in the two compared conditions of pixel i, respectively, N is the number of trials, and S$_i$ is the standard deviation of $(R_{i1} - R_{i2})$. Color preference maps were obtained by comparing red/green and white/black grating images, spatial frequency maps by comparing high and low spatial frequency images, and orientation maps by comparing two orthogonal orientation images (0° vs. 90° or 45° vs. 135°). For curvature maps, we compared the sum of curved gratings with the sum of straight gratings (four orientations: 0°, 45°, 90°, 135°). For single condition maps, we compared the selected condition with the averaged of straight gratings (four orientations: 0°, 45°, 90°, 135°). Maps were high-pass filtered (Gaussian filter, 10 pixel diameter) and low-frequency noise was reduced by convolving the map with a 100–150 pixel diameter circular filter and subtracted from the original maps. The orientation preference angle maps were calculated based on the single condition maps, and each pixels were assigned with a unique color to represent the preferred orientation (*Bosking et al., 1997*).

### Locating positions of functional domains

Functional domains were identified by selecting the pixels with significant dR/R difference (two-tailed t test, p<0.01) between two comparison conditions (color domains, red/green versus white/black, ΔdR/R < 0; high spatial frequency domains, high spatial frequency versus low spatial frequency, ΔdR/R < 0; 0° orientation domain, 0° versus 90°, ΔdR/R < 0; 45° orientation domain, 45° versus 135°, ΔdR/R < 0; 90° orientation domain, 90° versus 0°, ΔdR/R < 0; 135° orientation domain, 135°

versus 45°, ΔdR/R < 0; Curvature domains, curved gratings versus straight gratings, ΔdR/R < 0). In addition to t-test, pixels that belong to curvature domains were further assessed by one-way ANOVA (curved stimuli versus straight stimuli, $p<0.05$ with post-hoc Tukey Kramer correction) test. Domain size was calculated as size = $\pi R^2$, where area size is the size of each separated patch (pixel number ×area of each pixel), R is the average of long and short axis of patch. Patches smaller than 0.2 mm (diameter) were excluded from this analysis, as the reliability of these small patches is less secure. For each activated region, the geometrical center was calculated as its activation center (see *Figure 7*).

## Timecourse

For time courses, the value of each pixel was calculated first using the following functions: dR/R=(Fx-F0)/F0, where F0 is the average reflectance value of the first two frames (taken before visual stimuli onset), Fx is the reflectance value corresponding to frame X (X = 1, 2, 3...). To examine the response of a domain, we generated domain timecourses by averaging timecourse of all significant pixels within the domain (compared to blank, $p<0.01$, two-tailed t test; see *Figures 2* and *5–7*). To compare the responses of a domain to two stimulus conditions, we conducted a nonparametric test (Wilcoxon test, see *Figure 2L*) for values within frames 9–18. For some comparisons, the average of response groups were compared (e.g. all curved vs all straight).

## Similarity

To evaluate the similarity between two single condition maps, we extract the responses in the imaged area V4 (regions that could be activated by curvature stimuli during the experiment) and calculated the correlation coefficient values between the two response maps (see *Figures 3* and *4*). To evaluate the relationship between curvature degree difference and response similarity, we use the Regression function from Matlab. In this analysis, we use the condition number to represent the relative curvature degree of each curvature (see *Figure 3*). The curvature degree difference is based on the relative distance between two curvature degrees.

## Statistical analysis

For all statistical comparisons of functional maps, we use two-tailed t test or ANOVA. We used the Wilcoxon rank sum for the comparisons between response amplitudes in two conditions when data failed to show a normal distribution.

# Acknowledgements

This research was conducted at Zhejiang University and was supported by the National Key R & D Program of China (2018YFA0701400 to AWR), the National Natural Science Foundation of China (8191101288, 31627802, 81430010 to AWR), and the Fundamental Research Funds for the Central Universities (2019XZZX003-20 to AWR, 2020FZZX001-05 to SXM). No US funds (NIH, OHSU) were used to support these experiments or preparation of this manuscript.

# Additional information

## Funding

| Funder | Grant reference number | Author |
|---|---|---|
| National Key R & D Program of China | 2018YFA0701400 | Anna Wang Roe |
| National Natural Science Foundation of China | 8191101288 | Anna Wang Roe |
| National Natural Science Foundation of China | 31627802 | Anna Wang Roe |
| National Natural Science Foundation of China | 81430010 | Anna Wang Roe |
| The Fundamental Research | 2019XZZX003-20 | Anna Wang Roe |

Funds for the Central Universities

| The Fundamental Research Funds for the Central Universities | 2020FZZX001-05 | Xue Mei Song |

The funders had no role in study design, data collection and interpretation, or the decision to submit the work for publication.

## Author contributions

Jia Ming Hu, Data curation, Software, Formal analysis, Validation, Investigation, Visualization, Methodology, Writing - original draft, Writing - review and editing; Xue Mei Song, Data curation, Investigation, Writing - original draft; Qiannan Wang, Investigation; Anna Wang Roe, Conceptualization, Supervision, Funding acquisition, Validation, Writing - original draft, Project administration, Writing - review and editing

## Author ORCIDs

Jia Ming Hu https://orcid.org/0000-0002-5306-445X
Anna Wang Roe https://orcid.org/0000-0003-4146-9705

## Ethics

Animal experimentation: All procedures were performed in accordance with the National Institutes of Health Guidelines and were approved by the Zhejiang University Institutional Animal Care and Use Committee with the approved protocols (Permit Number:zju20160242).

## Decision letter and Author response

Decision letter https://doi.org/10.7554/eLife.57261.sa1
Author response https://doi.org/10.7554/eLife.57261.sa2

# Additional files

## Supplementary files

• Transparent reporting form

## Data availability

All data generated or analysed during this study are included in the manuscript and supporting files.

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
