## [Decision Letter]

**Acceptance summary:**

Curvature is a fundamental component of our ability to recognise and discriminate visual objects and therefore interact with the world around us. This optical imaging study compellingly demonstrates a systematic functional architecture of specialised domains for processing and representing curvature in the visual cortex of primates, in particular visual area V4. This finding has profound implications for our understanding of how primates perceive and recognise shape.

**Decision letter after peer review:**

Thank you for submitting your article "Curvature domains in V4 of Macaque Monkey" for consideration by *eLife*. Your article has been reviewed by three peer reviewers, and the evaluation has been overseen by a Reviewing Editor and Joshua Gold as the Senior Editor. The following individuals involved in review of your submission have agreed to reveal their identity: Aniruddha Das (Reviewer #2); Ed Connor (Reviewer #3).

The reviewers have discussed the reviews with one another and the Reviewing Editor has drafted this decision to help you prepare a revised submission.

As the editors have judged that your manuscript is of interest, but as described below that substantial additional analyses are required before it is published, we would like to draw your attention to changes in our revision policy that we have made in response to COVID-19 (https://elifesciences.org/articles/57162). First, because many researchers have temporarily lost access to the labs, we will give authors as much time as they need to submit revised manuscripts. We are also offering, if you choose, to post the manuscript to bioRxiv (if it is not already there) along with this decision letter and a formal designation that the manuscript is "in revision at *eLife*". Please let us know if you would like to pursue this option. (If your work is more suitable for medRxiv, you will need to post the preprint yourself, as the mechanisms for us to do so are still in development.)

Summary:

In this manuscript, Hu and colleagues investigate the functional organisation of important elements for representing object shape in the primate brain. Their focus of investigation is the mapping of curvature in extrastriate visual area V4, an area in which there is substantial evidence for neurons representing intermediate visual features that are more complex than the representations of single orientation in receptive fields of V1 and V2. The authors show that there are clearly regions of V4 that are activated better by curvature stimuli than traditional single orientation stimuli and thus provide potentially exciting and novel data about organization of curvature orientation in area V4. Curvature tuning is widely regarded as a major step toward complex shape perception that arises specifically in V4, the third stage in primate ventral pathway.

However, there are some major concerns about the scope and specificity of the presented data and data analyses as well as the discussion of the results, which currently limit the ability of the reader to fully appreciate to what extent the organisation for curvature: (a) is truly specialised across V4; (b) cannot be explained by end stopped cells, clustering of orientation-tuning or size tuning; and (c) is distinct from the neuronal properties found in earlier visual areas V1 and V2. In general, we suggest the authors present their entire dataset, use the same analyses across the full range of imaging data from V4 and potentially extend to V1 and V2, and use statistical measures to analyse the responses around curvature clusters and pin wheels to establish the existence of gradients for curvature parameters. More detailed comments and suggestions are presented below. We think that these revisions and additional analyses should bring out the evidence of a specialised, spatially organized representation of curvature more clearly.

Essential revisions:

1) Additional analyses to strengthen the case for specialised curvature organisation in V4.

In some parts of the paper (Figures 3 and 5), the authors rely on pixel-wise correlation as a proxy for mapping. But pixel correlation conveys no information about spatial organization, and high correlations are equally compatible with entirely random organization of signals. For example, any pattern (even a scramble) is likely to preferentially activate some neurons by chance and therefore some pattern of activation across the cortex (e.g. Figure 2—figure supplement 6C). If a different pattern (like another scrambled) is presented it is likely that a different pattern is obtained, and if I do any kind of morphing between these images, I am necessarily going to gradually shift the pattern of activation from one to the other. But this says little about the nature of representation. There is also an issue of map noise: within condition consistency is often below 0.7, and it's not clear why.

Only graphical analysis of images showing spatial clustering of similar tuning, spatial relationships between clusters, and/or spatial gradients of tuning changes are meaningful for understanding mapping, and this paper only gets to that stage in Figure 7, and even then only for one relatively small region of one case. The critical results shown there (and the corresponding but unanalyzed results in Figures 6A and S9A,B) are the only convincing evidence of organization within V4 curvature domains, and that organization in turn is the most convincing evidence that curvature per se is represented rather than something correlated with curvature.

Even these plots confound two phenomena. One phenomenon appears to be true clustering of curvature direction tuning. The strongest prediction for curvature direction tuning is that opposite directions (differing by 180 degrees) should be the most separated. This is clearest, surprisingly, in Figures S9A and B, where regions tuned for the up direction are largely complementary to regions tuned for down. This convinces me that the authors have observed organization for curvature direction, and this case in particular deserves to be featured in the main text and analyzed as in Figure 7.

In contrast, especially in Figures 6 and 7, there are many regions where the responses to opposite directions are largely overlapping, and instead it is directions that are 90 degrees different that are most complementary. In those cases, the same straight to curved gradients will be observed for two opposite directions (as is the case for the central three gradients in Figure 7C). The simplest interpretation of this is tuning for component orientations, because curved gratings, especially in the higher curvature domain, share component orientations across 180 degree differences, but have distinct orientations across 90 degree differences. A more complex interpretation, but one supported by previous literature, is that some domains imaged here represent contours with multiple curvatures. This would be consistent with findings that many V4 neurons represent contour fragments containing multiple curvature directions (Pasupathy and Connor, 2001; Carlson et al., 2011). These are frequently curvatures with opposite directions, since many contour fragments contain adjoining convex and concave curves, which out of geometric necessity tend to point in opposite directions. In this sense, the three gradients in the center of 7C, and similar gradients (not analyzed) in Figure 6 may reflect the geometric structure of the natural world.

Our strong suggestion would be to analyze all the available imaging regions across V4 (not just individual cases or smaller regions within cases) with the analyses in Figure 7, making clear distinctions between gradients that are specific to a single curvature direction and those that overlap for two curvature directions, and discussing possible interpretations of the latter case.

2) Additional analyses to elucidate the functional nature of curvature domains and the "pinwheel" structures.

The lack of single unit data here is problematic, especially given the authors' model, because a curvature "region", especially given its proximity to similar orientation regions, could simply be comprised of a mix of oriented cells with no individual cells showing curvature sensitivity. Looking at Figure 5, all curvatures are somewhat correlated with each other and with many orientations; this is clearly not the case for orientation in which there is basically no overlap between orientations. The model the authors present relies on the proximity of (and overlap between) curvature and orientation regions, but a "multi-orientation" region that includes individual neurons with a diversity of orientations tunings, but has no neurons with actual curvature preferences, would be equally plausible and consistent.

Because there is no single unit data, the regions of "curvature" activation could simply result from the co-activation of classically oriented cells (especially near pinwheel centers where such cells are likely to be colocalized). The authors are concerned that the presence of high spatial frequencies in the curvature stimuli is a confound, but there's no attempt to control for the increase in orientation content with curvature. Strong curvature contains a number of orientations, and we would therefore expect to activate multiple orientation columns, but this is not controlled for.

We believe it would be straightforward to control for: on a pixel-by-pixel basis see how responses to a curve could be explained by a weighted sum of the responses to the orientations that it contains. For example, small curvature responses should be similar to single orientation responses, while high curvature responses should be explainable by a mix of orientation responses (which according to map correlation data seems very reasonable). And an exact weighting of what orientations are present in a given curvature stimulus, can be done by looking at the 2-d Fourier transform of the stimulus.

Also, a similar analysis to that described in (point 1 above / Figure 7 of the manuscript) should be done to look for pinwheel structures, i.e. with overlapping contour maps and with gradient lines, and with clear presentation of the statistical significance of gradients presented prominently in the main paper. This may support the idea of pinwheel structures for curvature direction, in a way that would be graphically and statistically convincing. In contrast, the basis for the pinwheel maps in Figure 7D is unclear, so they are not appropriate for inclusion as presented. Even if pinwheels cannot be clearly identified, and if not all images show clear clustering for tuning, the main point of the paper would be established.

3) Extending analyses to available imaging data beyond V4 to strengthen the case that the organisation for curvature in V4 is different from functional representations in V1 and V2.

We were wondering why V1 and V2 were excluded. It looks like the authors are throwing out 2/3 of the data, and a vital component to making the case that there's something "special" going on in V4 is comparing it to other areas. On a similar note, V1 in case 2 looks problematic: there's no blob like activation of color and orientation columns aren't obvious (especially compared to the textbook examples shown in cases 1 and 3). Another exclusion issue: Figures 2 3, and 5 place letters so they obscure a substantial fraction of the imaged region.

A number of pertinent experimental details are also missing: there are 3 animals, but how many sessions, are there analyses in which the observations of multiple sessions are combined, and are there comparisons being made across sessions? Are stimuli presented binocularly? What are the RF locations of the V1, V2, and V4 regions?

Please extend the central analysis applied to V4 to V2 and V1 data and complete the requested information.

4) Discussing the results in the context of previous findings for single neuron tuning to show more clearly what other interpretations of the presented results are possible or can be rejected.

The curvature maps appear convincing: there are clearly regions of neurons that are activated better by curvature stimuli than traditional single orientation stimuli. However a simple explanation, dating back to 1965, of this would be the presence of end-stopped and surround suppression, and this is not addressed. The historical context seems a little off here; Livingstone's findings of 2017 are cited, but somehow Hubel and Wiesel, who in 1965 (J Neurophysiol) suggested that end-stopped, length tuned neurons could be useful for curvature detection, is omitted. There was a nice explicit model of how that could happen by Dobbins et al. in 1987 (Nature). But the manuscript makes no reference to any size tuning or surround suppression, and how that, in combination with classic orientation tuning, could create a cell that responds vigorously to curvature. This is even a greater concern given the well-established prevalence of size tuning, both electrophysiologically (1987) and with optical imaging (1997), in area V4. This has a huge impact on both novelty and interpretability; if you have a region of surround suppression (which we know exist) and it overlaps with an orientation region, there's a "curvature" region in that it will respond to a curve better than extended straight line or grating. But I would argue it's not a "curvature" region, since a short bar or grating without any curvature is actually the optimal stimulus. The absence of any reports of eccentricity or size tuning measurements does not install confidence that this is not what is responsible for these observations.

We think a clearer discussion of these issues would be important for the paper.

[Editors' note: further revisions were suggested prior to acceptance, as described below.]

Thank you for resubmitting your article "Curvature domains in V4 of Macaque Monkey" for consideration by *eLife*. Your revised article has been reviewed by two peer reviewers, and the evaluation has been overseen by a Reviewing Editor and Joshua Gold as the Senior Editor. The following individual involved in review of your submission has agreed to reveal their identity: Aniruddha Das (Reviewer #2).

The reviewers have discussed the reviews with one another and the Reviewing Editor has drafted this decision to help you prepare a revised submission.

We would like to draw your attention to changes in our revision policy that we have made in response to COVID-19 (https://elifesciences.org/articles/57162). Specifically, when editors judge that a submitted work as a whole belongs in *eLife* but that some conclusions require a modest amount of additional new analysis/data, as they do with your paper, we are asking that the manuscript be revised to either limit claims to those supported by data in hand, or to explicitly state that the relevant conclusions require additional supporting data.

Summary:

In this manuscript, Hu and colleagues investigate the functional organisation of important elements for representing object shape in the primate brain. Their focus of investigation is the mapping of curvature in extrastriate visual area V4, an area in which there is substantial evidence for neurons representing intermediate visual features that are more complex than the representations of single orientation in receptive fields of V1 and V2. The authors show that there are clearly regions of V4 that are activated better by curvature stimuli than traditional single orientation stimuli and thus provide potentially exciting and novel data about organization of curvature orientation in area V4. Curvature tuning is widely regarded as a major step toward complex shape perception that arises specifically in V4, the third stage in primate ventral pathway.

The manuscript is largely improved, and the authors have effectively addressed some of the major concerns raised in the initial submission. For instance, the authors addressed the concern that the curvature is fully accounted for by end stopping / surround suppression. They show that short straight lines poorly stimulate “curvature” regions and that “end stopped” patches are substantially distinct from “curvature” patches even though they overlap partially (Figure 2—figure supplement 13). Additionally, while showing that end stopping and curvature are not identical, the authors are careful in stating that end stopping could play an important role in curvature processing. But an important concern appears incompletely addressed. The evidence for the curvature specific domains is strong and novel, however the reviewers felt that the claims regarding the specialised architecture of these domains, namely the ordered, gradual representation of different degrees of curvature and the existence of pinwheels have not been fully supported by the presented results. We suggest either addressing the additional evidence suggested in revisions or tempering these claims and addressing these issues in a subsequent, linked publication.

1) One issue that is not satisfactorily addressed is “curvature pinwheels”. The authors make a point of proposing that pinwheel structure is fundamental to the organization of maps like orientation (in V1) and curvature (V4). While that proposal could arguably be made for V1 (Li, 2019) the authors' own evidence here suggests that pinwheels might be a red herring for curvature in V4.

a) The authors only show pinwheels in one data set (Figure 7B(right hand panel). And there too only the single pinwheel in the upper curvature patch looks reasonable. In the lower curvature patch the upper pinwheel lies very close to the edge of the curvature region making the classification problematic given the ~250-micron FWHM spatial resolution of intrinsic signal optical imaging (Polimeni, PNAS 2005).

b) More importantly, the most striking characteristic in the “curvature orientation” maps appears to be the high degree of overlap between regions of opposite curvature direction (Figure 6D). The net “curvature orientation” vector in these regions of overlap are presumably close to zero vector length. The maps (e.g. Figure 7B) with saturated colors are thus likely misleading as the real pattern likely involves not point singularities of “curvature orientation” (“pinwheels”) but large overlap patches of poor net “curvature orientation” with short “curvature orientation”-specific extensions outwards. Notably the authors did not show “curvature-orientation” contour maps, quantify the local orientation gradient and test it for significance to support their proposal of pinwheels. It is likely that the signal to noise in such maps would be very low.

c) In the Discussion, it may be worth considering if the pattern of large patches of overlap with orientation-specific extensions may be the dominant topological motif, rather than trying to squeeze the map into a straitjacket of pinwheels.

2) With respect to the gradual shift in curvature maps, the response to the point of of pixel-wise map correlation was not adequate: the authors' response states that it merely establishes a curvature map, but any superposition of single condition domains, or differential analysis between different curvatures, establishes this. The manuscript says something more: there is a shifting map, implying some sort of continuity of representation. As stated in the previous comments, this strikes us as poor logic: if we take a low frequency, chromatic, linear grating, at one extreme, and a high frequency achromatic curve at another, the two activation maps will look different, and if I do a pixel wise morphing between those two stimuli that doesn't know anything about high level attributes like color, curvature, and SF, I necessarily have to create activation maps that look intermediate. It may be that subtle non-continuities, like fractures and pinwheels, in functional maps will make the change in correlations non-linear, but I'm not even sure of that. In other words, this analysis has to succeed, and is therefore, at least as it is presented, uninformative. If the authors are interested in the regularity or periodicity of representations, that a correlation analysis of cortical distance vs representational distance, seems more appropriate. For example, the spatial correlation of curvature preference on a pixel-wise basis, or between "centers" of domains. But, on the basis of what is presented, we are not sure how that is actually going turn out. For example, Figure 7 seems to show that 3/4 of the center of masses of "up" and "down" curvatures are basically on top of one another.

Revisions expected in follow-up work:

The reviewers felt that the claims regarding the specialised architecture of these domains, namely the ordered, gradual representation of different degrees of curvature and the existence of pinwheels have not been fully supported by the presented results. We suggest either addressing the additional evidence suggested in revisions or tempering these claims and addressing these issues in a subsequent, linked publication. See previous section for details.

---

## [Author Response]

Essential revisions:1) Additional analyses to strengthen the case for specialised curvature organisation in V4.In some parts of the paper (Figures 3 and 5), the authors rely on pixel-wise correlation as a proxy for mapping. But pixel correlation conveys no information about spatial organization, and high correlations are equally compatible with entirely random organization of signals. For example, any pattern (even a scramble) is likely to preferentially activate some neurons by chance and therefore some pattern of activation across the cortex (e.g. Figure 2—figure supplement 6C). If a different pattern (like another scrambled) is presented it is likely that a different pattern is obtained, and if I do any kind of morphing between these images, I am necessarily going to gradually shift the pattern of activation from one to the other. But this says little about the nature of representation. There is also an issue of map noise: within condition consistency is often below 0.7, and it's not clear why.

Correlations between pairs of curvature stimuli:

1) 0.7 is not low. As shown in Figure 3—figure supplement 1, the comparisons of matched straight orientation maps, which are the best possible correlation indices, range from 0.66 to 0.9. In Figures 3 and 4 (previous Figures 3 and 5), the values for matched curvature maps, based on correlation between two conditions, also fall within this range (0.68-0.91). In our preparations, with the signal sizes recorded in V4 under anesthesia, and with data collected over hours of time, these values are typical. Figure 3—figure supplement 1 provides a benchmark by which to evaluate the value of the correlations. These values have now been incorporated in text.

2) Correlation values are related to curvature content. We agree with the reviewer that, in principle, correlations can result from non-curvature related responses, and even scrambled responses. However, here, we start with the observation that maps are differentially responsive to different degrees of curvature grating (curvature degree, Figure 3; curvature orientation, Figure 4). Then we show that these maps are systematically shifting using map correlations. We spend the rest of the manuscript providing additional support that these maps are truly curvature-specific responses. When considered in its entirety, our data supports the existence of curvature maps in V4.

Only graphical analysis of images showing spatial clustering of similar tuning, spatial relationships between clusters, and/or spatial gradients of tuning changes are meaningful for understanding mapping, and this paper only gets to that stage in Figure 7, and even then only for one relatively small region of one case. The critical results shown there (and the corresponding but unanalyzed results in Figures 6A and S9A,B) are the only convincing evidence of organization within V4 curvature domains, and that organization in turn is the most convincing evidence that curvature per se is represented rather than something correlated with curvature.

Paper organization: In response to the general critique that the maps for curvature domains come late in the paper, we think it is worthwhile to explain our general strategy. Our approach was to understand the organization of curvature response from global to local scale in V4. (1) We first show that response to curvature (preference for the curvature stimulus over the straight stimulus) falls within the “orientation bands” and not the “color” bands, and “curvature” domains are distinct from orientation domains. This finding alone is new (Figure 2). (2) We then examine, at the domain map scale, the organization and relationship with straight orientation maps. This provided some support for the possibility that responses might be related to curvature degree (Figures 3: Case 1, S11: Case 2, shifting correlations related to curvature degree) or to curvature orientation (new Figures 4: Case 2, S11: Case 1, higher when matched). Based on these global signals, V4 can distinguish different curvature degrees and curvature orientations. (3) To further examine whether these correlations are due to the curvature content, we analyzed the topographic region of V4 corresponding to the central (~1 degree) of the curvature stimulus (total stimulus size is 4 deg), which is the part with high curvature content. This revealed substructure within the putative curvature domains with preferential response for curvature degree and orientation (Figures 5, 6). (4) Finally, we made an attempt to provide some hypothesis about the global organization of curvature representation (Figure 7G). We concluded that, when taken all together, our data are consistent with the hypothesis that there is a systematic “curvature map” within V4. This is the first study showing such systematic maps at domain scale (cf. Yue et al., 2014, showed large regions responsive to curvature but did not report maps for curvature degree or curvature orientation. Tang et al: we recently learned of this study under review at e*Life* and have cited this study).This strategy is now described in the beginning of the Results under “General Approach”.

Even these plots confound two phenomena. One phenomenon appears to be true clustering of curvature direction tuning. The strongest prediction for curvature direction tuning is that opposite directions (differing by 180 degrees) should be the most separated. This is clearest, surprisingly, in Figures S9A and B, where regions tuned for the up direction are largely complementary to regions tuned for down. This convinces me that the authors have observed organization for curvature direction, and this case in particular deserves to be featured in the main text and analyzed as in Figure 7.

Adding Figure S9 to main text: complementary representation of up/down orientation: As the reviewers suggest, we have now added Figure S9 to Figure 6 (the new Figure 6A-C), and describe it in the main text. Figure 6A illustrates a region where opposing orientations (illustrated with both curved gratings and curved lines) are largely non overlapping. Figure 6A-C is now described in the text.

In contrast, especially in Figures 6 and 7, there are many regions where the responses to opposite directions are largely overlapping, and instead it is directions that are 90 degrees different that are most complementary. In those cases, the same straight to curved gradients will be observed for two opposite directions (as is the case for the central three gradients in Figure 7C). The simplest interpretation of this is tuning for component orientations, because curved gratings, especially in the higher curvature domain, share component orientations across 180 degree differences, but have distinct orientations across 90 degree differences. A more complex interpretation, but one supported by previous literature, is that some domains imaged here represent contours with multiple curvatures. This would be consistent with findings that many V4 neurons represent contour fragments containing multiple curvature directions (Pasupathy and Connor, 2001; Carlson et al., 2011). These are frequently curvatures with opposite directions, since many contour fragments contain adjoining convex and concave curves, which out of geometric necessity tend to point in opposite directions. In this sense, the three gradients in the center of 7C, and similar gradients (not analyzed) in Figure 6 may reflect the geometric structure of the natural world.Curvature domain response not due to component orientations. Our evidence suggests these curvature responses are difficult to attribute simply to component orientations. We test this by examining whether curvature domains exhibit strong response to the 0,45,135 deg components of the curvature stimulus. In Figure 7—figure supplement 2A-D, we show that, while there are locations with robust response to straight stimuli (red lines in B,C: 0,45,135 graphs), the curvature domains exhibit little response to component straight stimuli. Instead, they show strong response only to curvatures; these responses are graded with changing curvature degree (dark to light gray lines). This suggests that the curvature response is not primarily a response to components of the curve. We also show (see “Not due to weighted sum” below) that the curvature response is not a weighted sum of straight responses. We have added text in Results under “Curvature domains are not pinwheel centers or sums of orientation components”.

Some V4 domains may represent contours with multiple curvatures or other invariant responses.

As the reviewer states, V4 neurons can represent contour fragments containing multiple curvature orientations (Pasupathy and Connor, 2001; Carlson et al., 2011). In this paper, we have taken an initial look at the organizations related to simple curvature degree and curvature orientation. We do not understand nor have we explored other possible combinations of different curvature response that may exist in V4. We believe that there may be higher order integrations reflecting combinations of curvatures or cue-invariant curvature response or even simple shapes (cf Tang et al. submitted). This parallels our thinking in V2, where there may be distinct domains responsive to single cue (e.g. motion dot borders) vs invariant multi-cue (e.g. regardless of which type of cue) border domains (Ramsden et al., 2015). Such possibilities will require further study.

There are indications within our data of possible higher order responses that are activated by different combinations of curvature orientations. For example, we illustrate locations where pixels are responsive to both upwards and downwards or both leftwards and rightwards curvature orientations (new Figure 6D brown regions). The percentage of such overlapping pixels is relatively small (overlapping pixels vs. all colored pixels was on average 22.4%; Case 1: 32.1%; Case 2: 14.9%; Case 3: 20.3%). Thus, although there were “complex” pixels responsive to multiple orientations, representation of opposing curvature orientations was mostly non-overlapping.

In sum, we believe that the organization is likely to contain further complexity and requires further study. We indeed embrace the idea that representation of convex and concave “shape fragments” may derive from statistics of natural stimuli (e.g. David et al., 2004 J Neurosci; Movshon and Simoncelli, 2015). Hebbian-based principles may lead to establishment of systematic domain-based maps that reflect statistics of the natural environment. Such domain diversity would be consistent with a mid-tier stage of shape representation.

Our strong suggestion would be to analyze all the available imaging regions across V4 (not just individual cases or smaller regions within cases) with the analyses in Figure 7, making clear distinctions between gradients that are specific to a single curvature direction and those that overlap for two curvature directions, and discussing possible interpretations of the latter case.

As we mentioned above, our strategy was to focus on the region responsive to the central curvature part of our stimulus. Figures 4-7 are focused on this central representation. As shown in Figure 7—figure supplement 2A, we purposely focused on the central region (~1 deg, red box) of the 4 deg curvature stimulus. This is the region that contains the greatest changing curvature across curvature degree stimuli. In addition, the central region experiences relatively constant curvature orientation and spatial frequency across conditions, unlike the flanking parts of the stimulus (blue box), which contain large spatial frequency and orientation differences across conditions). Thus, the non-central regions are distinct in that they contain relatively straight components and a broader mixture of orientations and spatial frequencies across conditions. This has been clarified in Results under “General Approach” and “the central ~1 deg representation” stated where appropriate in the text.

V1/V2 and V4 represent different retinotopic locations in the same field of view. In each experiment, we identified cortical location corresponding to central ~1 deg of the stimulus. To determine the region of cortex corresponding to this central region, we first mapped, monocularly, the RF location of the V4 region (Figure 2—figure supplement 2), using vertical and horizontal 0.2° wide bars (vertical: different x; horizontal: different y, Figure 2—figure supplement 2A,G,M). In each case, the central ~1 deg stimulus covered a cortical area roughly ~4-5mm in diameter; we therefore evaluated curvature response from this small region. In some experiment sessions, we also presented a flashed white square (size=1°) at the determined V4 center (X0, Y0) to estimate the coverage area of the central part of the stimulus (See Figure 2—figure supplement 2H and M). This is now explicitly stated in text and retinotopic mapping methods described in text and Materials and methods under “Retinotopic mapping”.

2) Additional analyses to elucidate the functional nature of curvature domains and the "pinwheel" structures.The lack of single unit data here is problematic, especially given the authors' model, because a curvature "region", especially given its proximity to similar orientation regions, could simply be comprised of a mix of oriented cells with no individual cells showing curvature sensitivity. Looking at Figure 5, all curvatures are somewhat correlated with each other and with many orientations; this is clearly not the case for orientation in which there is basically no overlap between orientations. The model the authors present relies on the proximity of (and overlap between) curvature and orientation regions, but a "multi-orientation" region that includes individual neurons with a diversity of orientations tunings, but has no neurons with actual curvature preferences, would be equally plausible and consistent.Because there is no single unit data, the regions of "curvature" activation could simply result from the co-activation of classically oriented cells (especially near pinwheel centers where such cells are likely to be colocalized). The authors are concerned that the presence of high spatial frequencies in the curvature stimuli is a confound, but there's no attempt to control for the increase in orientation content with curvature. Strong curvature contains a number of orientations, and we would therefore expect to activate multiple orientation columns, but this is not controlled for.

Curvature domains are not pinwheel centers. The reviewer raises a good point, concerning the possibility that curvature responses could be due to activation of a mixture of oriented responses, such as that found at orientation pinwheel centers. (1) As shown in the following figure (new Figure 2—figure supplement 12), the curvature domains (orange regions) do not co-localize with orientation pinwheel centers (blue dots). In fact, most of the pinwheel centers are outside the curvature domains entirely. (2) Moreover, as shown by the responses of pinwheel centers to straight and curved stimuli (E), the pinwheel center responses to curvatures are weak and do not distinguish curvature from straight (Wilcoxon test, p=0.13). This makes it unlikely for pinwheel centers to be locations of curvature response. We have added text in Results under “Curvature domains are not pinwheel centers or sums of orientation components”.

Lack of electrophysiology: Given the time-consuming nature of our optical imaging data collection, we were not able to focus on electrophysiological characterization. As the goal of this study was to evaluate the functional organization of V4, we focused our efforts on collecting optical imaging data. However, we did collect a small number of units. During imaging of Case 3, we targeted a few penetrations to curvature domains (Figure 2—figure supplement 9). Although the number of neurons recorded is small, our data are consistent with the curvature maps. We found that single-unit and multi-unit responses (n=26) in curvature domains (n=3) preferred curvature stimuli over straight stimuli and exhibited weak responses to straight stimuli. This suggests that curvature domains are not simply comprised of a mix of orientation cells. This is now described in the text.

We believe it would be straightforward to control for: on a pixel-by-pixel basis see how responses to a curve could be explained by a weighted sum of the responses to the orientations that it contains. For example, small curvature responses should be similar to single orientation responses, while high curvature responses should be explainable by a mix of orientation responses (which according to map correlation data seems very reasonable). And an exact weighting of what orientations are present in a given curvature stimulus, can be done by looking at the 2-d Fourier transform of the stimulus.

Not due to a weighted sum. According to reviewers’ comments, we show the cortical responses to the orientations in different curvature stimuli. As shown in the Figures 7—figure supplement 2B-D, the stimuli with low curvature degrees tend to be similar to the single orientation responses. However, for high curvature degrees, the responses of orientation domains were weak, only curvature domains show strong response. Thus the high curvature responses in curvature domains cannot be explained simply by a weighted sum of these orientation responses (2D fourier transform revealed that there are orientation components around 45° and 135° in the curvature stimuli, Figure 7—figure supplement 2A). To do a quick calculation of a weighted sum of these orientation responses, we chose the responses of 0, 45, 135 orientation domains (as these domains cover the range of the orientations in the curvature stimuli). , here Response is the weighted sum of these orientation responses, Response_i_ represents the response of corresponding orientation domain (i=0, 45, 135) to curvature, Wi represents the weight of different orientations, here we consider these orientations contribute equally, thus Wi=1/3. For upwards curvature (from Figure 7—figure supplement 2C, a/b ratio=7), Response_0_=-0.0026%, Response_45_=-0.0027%, Response_135_=0.0006%, Wi=1/3, Response=-0.0016%, much weaker (by almost an order of magnitude) than the response in curvature domains -0.0115% (responses were calculated by average of the response amplitudes from the last 10 frames). While we are not able to test the weighted sum of all possible orientations, at least with this simple test, we do not see any indication of component summation. We have added text in Results under “Curvature domains are not pinwheel centers or sums of orientation components” and in Discussion.

In summary: We focused on the regions corresponding to the central ~1 deg of the stimuli (where curvature content changes the most across stimuli). (1) We find that the curvature responses are not due to the component orientation signals, as these are located in the orientation domains. In domains with response to high curvature, the responses to orientation are very weak (Figure 7—figure supplement 2). (2) Orientation pinwheel centers and curvature domains are in different spatial locations (Figure 2—figure supplement 12). Orientation pinwheel centers and curvature domains had different response preferences: pinwheels failed to distinguish between curvature and straight, and their responses to curvatures are weak (Figure 2—figure supplement 9). (3) Electrophysiological recordings in curvature domains show that, at least in the small sample of penetrations, neurons predominantly prefer curvature. Thus the responses here are more likely due to the curvature content rather than orientation component features. Substantial text has been added in Discussion under “Testing other possible interpretations”.

Also, a similar analysis to that described in (point 1 above / Figure 7 of the manuscript) should be done to look for pinwheel structures, i.e. with overlapping contour maps and with gradient lines, and with clear presentation of the statistical significance of gradients presented prominently in the main paper. This may support the idea of pinwheel structures for curvature direction, in a way that would be graphically and statistically convincing. In contrast, the basis for the pinwheel maps in Figure 7D is unclear, so they are not appropriate for inclusion as presented. Even if pinwheels cannot be clearly identified, and if not all images show clear clustering for tuning, the main point of the paper would be established.

We agree that the data regarding potential pinwheels is not sufficient and have now removed any mention of curvature pinwheels from the paper.

3) Extending analyses to available imaging data beyond V4 to strengthen the case that the organisation for curvature in V4 is different from functional representations in V1 and V2.We were wondering why V1 and V2 were excluded. It looks like the authors are throwing out 2/3 of the data, and a vital component to making the case that there's something "special" going on in V4 is comparing it to other areas. On a similar note, V1 in case 2 looks problematic: there's no blob like activation of color and orientation columns aren't obvious (especially compared to the textbook examples shown in cases 1 and 3). Another exclusion issue: Figures 2 3, and 5 place letters so they obscure a substantial fraction of the imaged region.

Clear V1 responses in Case 2. The reviewers ask about the unclear responses in V1 from Case 2 (Figure 2—figure supplement 1D, E). In this case, the cortex underneath the coverglass was not flat, resulting in a band of cortex that was out of the plane of focus. However, distinct blob maps (Figure 2—figure supplement 1D) and some orientation maps (Figure 2—figure supplement 1E) were obtained in V1. In other sessions, we obtained strong V1 response in this Case (see Author response image 1).

**Author response image 1. sa2fig1:** Clear functional maps of Case 2 from a different experiment session. A. Color map. B. Orientation map.

A number of pertinent experimental details are also missing: there are 3 animals, but how many sessions, are there analyses in which the observations of multiple sessions are combined, and are there comparisons being made across sessions? Are stimuli presented binocularly? What are the RF locations of the V1, V2, and V4 regions?

Missing Details. (1) We did 27 experimental sessions on three hemispheres of two monkeys. Text has been added. We conducted 6-12 sessions in each hemisphere. Each experimental session was focused on answering different questions. (2) We did obtain repeats of some maps. For example, in Figure 2—figure supplement 5, we compared the curvature maps in two different experiment sessions from the same hemisphere. The maps are consistent across time. Text has been added. (3) As we used vecuronium bromide during the experiment, the two eyes did not look at the same position of the monitor, thus for curvature mapping, we covered one eye and only presented the curvature stimuli to one eye. Described in Materials and methods. (4) In every experiment session, we first mapped the RF location of V4 region and presented the stimuli at that location (~1-5 deg eccentricity in V4). For V1, locations were at about 1-2 deg eccentricity (see Figure 2 — figure supplement1). We did not have much V2 available on the surface in these cases (see Figure 2—figure supplement 11). We have now incorporate this into the Materials and methods.

Please extend the central analysis applied to V4 to V2 and V1 data and complete the requested information.

V2 not available on surface. In these three cases, there is limited amount of V2 on the surface; most of V2 is buried within the lunate sulcus (Figure 2—figure supplement 11). Thus we do not have data from V2. Text has been added.

Most of V1 not imaged due to retinotopic mismatch: See Figure 2—figure supplement 2.

V1 does not contain curvature domains. We examined whether V1 contains curvature maps. In two experimental sessions from Case 3 (see Figure 2—figure supplement 10), we positioned the stimulus center over the exposed V1. We first determined the visuotopic location by mapping a 1 deg stimulus (Figure 2—figure supplement 10A, red dotted line, position of stimulus differs by ~1 deg in Experiment 1 and Experiment 2). In both experimental sessions, when the 4 deg curvature stimulus was centered over this location, V1 failed to show any significant preference for curvature over straight (Figure 2—figure supplement 10B). So, this suggests that, unlike V4, V1 does not contain curvature domains; this finding is consistent with previous studies (Yue et al., 2014; Ponce et al., 2017). Text has been added.

4) Discussing the results in the context of previous findings for single neuron tuning to show more clearly what other interpretations of the presented results are possible or can be rejected.The curvature maps appear convincing: there are clearly regions of neurons that are activated better by curvature stimuli than traditional single orientation stimuli. However a simple explanation, dating back to 1965, of this would be the presence of end-stopped and surround suppression, and this is not addressed. The historical context seems a little off here; Livingstone's findings of 2017 are cited, but somehow Hubel and Wiesel, who in 1965 (J Neurophysiol) suggested that end-stopped, length tuned neurons could be useful for curvature detection, is omitted. There was a nice explicit model of how that could happen by Dobbins et al. in 1987 (Nature). But the manuscript makes no reference to any size tuning or surround suppression, and how that, in combination with classic orientation tuning, could create a cell that responds vigorously to curvature. This is even a greater concern given the well-established prevalence of size tuning, both electrophysiologically (1987) and with optical imaging (1997), in area V4. This has a huge impact on both novelty and interpretability; if you have a region of surround suppression (which we know exist) and it overlaps with an orientation region, there's a "curvature" region in that it will respond to a curve better than extended straight line or grating. But I would argue it's not a "curvature" region, since a short bar or grating without any curvature is actually the optimal stimulus. The absence of any reports of eccentricity or size tuning measurements does not install confidence that this is not what is responsible for these observations.

End-stopping and surround suppression are not sufficient to explain all the curvature responses. We appreciate the reviewer for raising the important question of whether size tuning may contribute to curvature preference response. As previous studies suggested, size tuning or end stopping may play an important role in curvature detection (Hubel and Wiesel, 1965; Dobbins et al., 1987) and is not independent from curvature preference features (Ponce et al., 2017). However other studies have suggested neurons with complex receptive field structures may also account for curvature preferences (Nandy et al., 2013, 2016). For this reason, while end-stopping likely contributes to curvature response, there are also other relevant parameters contributing to curvature response. (1) Surround suppressed and curvature domains are not the same: Similar to previous studies (Ghose and Ts'o, 1997), we found there were size sensitive regions in V4 (Case 2, Figure 2—figure supplement 13A). However, while there was some overlap, the size sensitive regions were on the whole spatially distinct from the curvature domains (compare Figure 2—figure supplement 13A and B, overlay in Figure 2—figure supplement 13C). (2) Weak response to small stimuli. Responses of curvature domains to small stimuli were weak in comparison to response to curvature stimuli (Figure 2—figure supplement 13D), suggesting weak end-stopping. Therefore curvature response cannot be fully explained by strong responses to end-stopping. (3) In addition, if end-stopping was the only factor that matters, curvatures with opposite orientations would densely overlap, which is not the case (Figure 6). Thus curvature preference response in V4 is not due to end-stopping alone. This is now described in Results under “Curvature domains are not end-stopping domains” and in Discussion under “Unlikely to be end-stopping domains”.

The curvature domain is based on curvature preference, not on surround suppression or end-stopping. The reviewer suggests that the imaged response to a curve may actually be a response to a short bar. We show in Figure 2—figure supplement 13 that this is not the case. The response to the small stimulus (“short bar”) is not strongly relative to the curvature response. While it is possible that some “curvature domains” may be termed “end-stopped domains” (i.e. regions of overlap in Figure 2—figure supplement 13C, 33.3% of orange pixels), at least some curvature domains exhibit poor end-stopping and are consistent with “curvature domain” (66.7% of orange pixels).

We think a clearer discussion of these issues would be important for the paper.

We have now completely rewritten this section and provided a more in depth discussion. It now includes a more in depth summary and the hypercolumn model (Figure 7E) has been moved to discussion. A new section entitled “Testing other possible interpretations” discusses possible contributions from component orientation, high spatial frequency, and end-stopping responses. Relationship of our findings with other studies is now expanded. In addition, we describe the significance of this work as providing a columnar hierarchy for shape representation.

[Editors' note: further revisions were suggested prior to acceptance, as described below.]

Revisions for this paper:The manuscript is largely improved, and the authors have effectively addressed some of the major concerns raised in the initial submission. For instance, ehe authors addressed the concern that the curvature is fully accounted for by end stopping / surround suppression. They show that short straight lines poorly stimulate “curvature” regions and that “end stopped” patches are substantially distinct from “curvature” patches even though they overlap partially (Figure 2—figure supplement 13). Additionally, while showing that end stopping and curvature are not identical, the authors are careful in stating that end stopping could play an important role in curvature processing. But an important concern appears incompletely addressed. The evidence for the curvature specific domains is strong and novel, however the reviewers felt that the claims regarding the specialised architecture of these domains, namely the ordered, gradual representation of different degrees of curvature and the existence of pinwheels have not been fully supported by the presented results. We suggest either addressing the additional evidence suggested in revisions or tempering these claims and addressing these issues in a subsequent, linked publication.1) One issue that is not satisfactorily addressed is “curvature pinwheels”. The authors make a point of proposing that pinwheel structure is fundamental to the organization of maps like orientation (in V1) and curvature (V4). While that proposal could arguably be made for V1 (Li, 2019) the authors' own evidence here suggests that pinwheels might be a red herring for curvature in V4.a) The authors only show pinwheels in one data set (Figure 7B(right hand panel). And there too only the single pinwheel in the upper curvature patch looks reasonable. In the lower curvature patch the upper pinwheel lies very close to the edge of the curvature region making the classification problematic given the ~250-micron FWHM spatial resolution of intrinsic signal optical imaging (Polimeni, PNAS 2005).

We agree that the data regarding potential pinwheels is not sufficient and have now removed any mention of pinwheels from the paper.

b) More importantly, the most striking characteristic in the “curvature orientation” maps appears to be the high degree of overlap between regions of opposite curvature direction (Figure 6D). The net “curvature orientation” vector in these regions of overlap are presumably close to zero vector length. The maps (e.g. Figure 7B) with saturated colors are thus likely misleading as the real pattern likely involves not point singularities of “curvature orientation” (“pinwheels”) but large overlap patches of poor net “curvature orientation” with short “curvature orientation”-specific extensions outwards. Notably the authors did not show “curvature-orientation” contour maps, quantify the local orientation gradient and test it for significance to support their proposal of pinwheels. It is likely that the signal to noise in such maps would be very low.

Yes, we agree with the reviewer’s assessment. We removed Figure 7A-D and revised the summary figure in Figure 7E.

c) In the Discussion, it may be worth considering if the pattern of large patches of overlap with orientation-specific extensions may be the dominant topological motif, rather than trying to squeeze the map into a straitjacket of pinwheels.

Yes, we agree. We have simplified the model (Figure 7E) to reflect the data that we present. These include basic curvature domains, the presence of curvature degree progressions, and potential highly overlapped complex curvature domains.

2) With respect to the gradual shift in curvature maps, the response to the point of of pixel-wise map correlation was not adequate: the authors' response states that it merely establishes a curvature map, but any superposition of single condition domains, or differential analysis between different curvatures, establishes this. The manuscript says something more: there is a shifting map, implying some sort of continuity of representation. As stated in the previous comments, this strikes us as poor logic: if we take a low frequency, chromatic, linear grating, at one extreme, and a high frequency achromatic curve at another, the two activation maps will look different, and if I do a pixel wise morphing between those two stimuli that doesn't know anything about high level attributes like color, curvature, and SF, I necessarily have to create activation maps that look intermediate. It may be that subtle non-continuities, like fractures and pinwheels, in functional maps will make the change in correlations non-linear, but I'm not even sure of that. In other words, this analysis has to succeed, and is therefore, at least as it is presented, uninformative. If the authors are interested in the regularity or periodicity of representations, that a correlation analysis of cortical distance vs representational distance, seems more appropriate. For example, the spatial correlation of curvature preference on a pixel-wise basis, or between "centers" of domains. But, on the basis of what is presented, we are not sure how that is actually going turn out. For example, Figure 7 seems to show that 3/4 of the center of masses of "up" and "down" curvatures are basically on top of one another.

We agree with the reviewer that a simple correlation analysis as presented in not sufficient to infer a systematic progression of shifting curvature domains. We have now added, as the reviewer suggested, analyses of the distances in Figure 7 (Case 1) and in Figure 7—figure supplement 1 (Cases 2 and 3). For each case, we focused on the regions that corresponded to the central ~1 deg of the stimulus; we analyzed all the straight domains and the corresponding curvature domains (with same orientation). Starting from a straight orientation domain, we selected the nearest domain of each adjacent condition (shown in Figure 7B) and measured the distances between center of mass of each domain and that of the straight domain (leftmost panel in Figure 7C). These values are plotted in Figure 7D. This illustrates that the curvature preference domains generally progress from low to high curvature with distance across the cortex. It is informative that the progression is not necessarily linear, a point we bring up in Discussion. We also do not suggest that this indicates any continuity of representation, merely that there is a generally increasing distance with curvature change. We thank the reviewer for suggesting this analysis. We also add in the legends of Figure 3 that the fitted lines are not meant to indicate linear fits; rather, they help the reader see the trends of the different groups of points. Analyses of the two other cases are shown in Figure 7—figure supplement 1.

In addition, we also observed examples of a clear separation between straight response and curvature response (Figure 7—figure supplement 1C) and domains of broad curvature degree preference (Figure 7—figure supplement 1G). Thus, not all curvature domains are organized in progressions.

In total across the 3 cases, on average, there is a general overall tendency for straight-to-curved domains to exhibit spatial shifts. However, there are some domains with broad curvature preference. This suggests the presence of a diversity in organization, indicating that curvature representation in V4 is complex.